# Diffused Redundancy in Pre-trained Representations

**Vedant Nanda** *
University of Maryland &
MPI-SWS

**Till Speicher**
MPI-SWS

**John P. Dickerson**
University of Maryland

**Krishna P. Gummadi**
MPI-SWS

**Soheil Feizi**
University of Maryland

**Adrian Weller**
The Alan Turing Institute
& University of Cambridge

## Abstract

Representations learned by pre-training a neural network on a large dataset are increasingly used successfully to perform a variety of downstream tasks. In this work, we take a closer look at how features are encoded in such pre-trained representations. We find that learned representations in a given layer exhibit a degree of *diffuse redundancy*, *i.e.*, any randomly chosen subset of neurons in the layer that is larger than a threshold size shares a large degree of similarity with the full layer and is able to perform similarly as the whole layer on a variety of downstream tasks. For example, a linear probe trained on $20\%$ of randomly picked neurons from the penultimate layer of a ResNet50 pre-trained on ImageNet1k achieves an accuracy within $5\%$ of a linear probe trained on the full layer of neurons for downstream CIFAR10 classification. We conduct experiments on different neural architectures (including CNNs and Transformers) pre-trained on both ImageNet1k and ImageNet21k and evaluate a variety of downstream tasks taken from the VTAB benchmark. We find that the loss & dataset used during pre-training largely govern the degree of diffuse redundancy and the "critical mass" of neurons needed often depends on the downstream task, suggesting that there is a task-inherent redundancy-performance Pareto frontier. Our findings shed light on the nature of representations learned by pre-trained deep neural networks and suggest that entire layers might not be necessary to perform many downstream tasks. We investigate the potential for exploiting this redundancy to achieve efficient generalization for downstream tasks and also draw caution to certain possible unintended consequences. Our code is available at https://github.com/nvedant07/diffused-redundancy.

## 1 Introduction

Over the years, many architectures have been proposed (such as [50, 16, 23]) that achieve competitive accuracies on many benchmarks [45]. A key reason for the success of these models is their ability to learn useful representations of data [28]. While these models continue to get better, understanding the properties of underlying learned representations continues to be a challenge.

Prior works have attempted to understand representations learned by deep neural networks through the lens of mutual information between the representations, inputs, and outputs [49] and hypothesize that neural networks perform well because of a "compression" phase where mutual information between inputs and representations decreases. Additionally, recent works have found that many neurons are *polysemantic*, *i.e.*, one neuron can encode multiple "concepts" [10, 39], and that one can

---

*Correspondence to: vnanda@mpi-sws.org

Table 1: Different model architectures with varying penultimate layer lengths trained on ImageNet1k. WRN50-2 stands for WideResNet50-2. Implementation of architectures is taken from `timm` [59]. Diffused redundancy here measures what fractions of neurons (randomly picked) can be discarded to achieve within $\delta = 90\%$ performance of the full layer.

| Model | Feature Length | ImageNet1k Top-1 Accuracy | Diffused Redundancy for $\delta = 0.9$ | | | |
|---|---|---|---|---|---|---|
| | | | CIFAR10 | CIFAR100 | Flowers | Oxford-IIIT-Pets |
| ViT S-16 | 384 | 64.82% | 0.70 | 0.50 | 0.50 | 0.80 |
| ViT S-32 | 384 | 55.73% | 0.70 | 0.50 | 0.50 | 0.70 |
| ResNet18 | 512 | 69.23% | 0.80 | 0.50 | 0.50 | 0.90 |
| ResNet50 | 2048 | 80.07% | 0.90 | 0.50 | 0.20 | 0.90 |
| WRN50-2 | 2048 | 77.00% | 0.95 | 0.80 | 0.50 | 0.95 |
| VGG16 | 4096 | 73.36% | 0.95 | 0.80 | 0.80 | 0.95 |

then train sparse linear models on such concepts to do "explainable" classification [61]. However, it is not well understood if or how extracted features are concentrated or spread across the entire representation.

While the length of the feature vectors extracted from state-of-the-art networks [2] can vary greatly, their accuracy on downstream tasks are not correlated to the size of the representation (see Table 1), but rather depend mostly on the inductive biases and training recipes [60, 53]. In all cases, the size of extracted feature vector (*i.e.* number of neurons) is orders of magnitude less than the dimensions of the input and thus allows for efficient transfer to many downstream tasks [22, 4, 41, 55]. We show that even using a *random* subset of these extracted neurons is enough to achieve downstream transfer accuracy close to that achieved by the full layer, thus showing that learned representations exhibit a degree of *diffused redundancy* (Table 1).

Early works in perception suggest that there are many redundant neurons in the human visual cortex [3] and some later works argued that a similar redundancy in artificial neural networks should help in faster convergence [19]. In this paper, we revisit redundancy in the context of modern DNN architectures that are trained on large-scale datasets. In particular, we propose the **diffused redundancy** hypothesis and systematically measure its prevalence across different pre-training datasets, losses, model architectures, and downstream tasks. We also show how this kind of redundancy can be exploited to obtain desirable properties such as generalization performance but at the same time draw caution to certain drawbacks of such an approach in increasing disparity in inter-class performance. We highlight the following contributions:

- We present the diffused redundancy hypothesis which states that learned representations exhibit redundancy that is diffused throughout the layer, *i.e.*, many *random subsets* (of sufficient size) of neurons can perform as well as the entire layer. Our work aims to better understand the nature of representations learned by DNNs.

- We present an initial analysis of why diffused redundancy exists in deep neural networks, we show that a randomly chosen subset of neurons of size $k$ performs as well as the projection of the entire layer on the first $k$ principal components. Intuitively this means that in DNNs' representations, *many* random subsets of size $k$ roughly capture all the variation in data that is possible with $k$ dimensions (since PCA represents directions of maximum variance).

- We propose a measure of diffused redundancy and systematically test our hypothesis across various architectures, pre-training datasets & losses, and downstream tasks.

  - We find that diffused redundancy is significantly impacted by pre-training datasets & loss and downstream datasets.
  - We find that models that are explicitly trained such that particular parts of the full representation perform as well as the full layer, *i.e.*, these models have *structured redundancy* (*e.g.* [26]), also exhibit a significant amount of diffused redundancy. Further, we also evaluate models trained with regularization that decorrelates activation of neurons and again find that these regularizations surprisingly *do not* affect diffused redundancy. These results suggest that this phenomenon is perhaps inevitable when DNNs have a wide enough final layer.

---

[2]Extracted features for the purpose of this paper refers to the representation recorded on the penultimate layer, but the larger concept applies to any layer

– We quantify the degree of diffused redundancy as a function of the number of neurons in a given layer. As we reduce the dimension of the extracted feature vector and re-train the model, the degree of diffused redundancy decreases significantly, implying that diffused redundancy only appears when the layer is wide enough to accommodate redundancy.

- Finally we draw caution to some potential undesirable side-effects of exploiting diffused redundancy for efficient transfer learning that have implications for fairness.

## 1.1 Related Work

Closest to our work is that of Dalvi et al. [7] who also investigate neuron redundancy but in the context of pre-trained language models. They analyze two language models and find that they can achieve good downstream performance with a significantly smaller subset of neurons. However, there are two key differences to our work. First, their analysis of neuron redundancy uses neurons from all layers (by concatenating each layer), whereas we show that such redundancy exists even at the level of a single (penultimate) layer. Second, and perhaps more importantly, they use feature selection to choose the subset of neurons, whereas we show that features are diffused throughout and that even a *random* pick of neurons suffices. Our work also differs by analyzing vision models (instead of language models) and using a diverse set of 30 pre-trained models (as opposed to testing only two models) which allows us to better understand the causes of such redundancy.

**Efficient Representation Learning** These works aim to learn representations which are "slim", with the goal of efficient deployment on edge devices [65, 64, 5]. Recently proposed paradigm of *Matryoshka Representation Learning* [26] aims to learn nested representations where one can perform downstream tasks with only a small portion of the representation. The goal of such representations is to allow quick, adaptive deployment without having to perform multiple, often expensive, forward passes. These works could be seen as inducing *structured redundancy* on the learned representations, where pre-specified parts of the representation are made to perform similar to the full representation. Our work, instead, aims to look at *diffused redundancy* that arises naturally in the training of DNNs. We carefully highlight the tradeoffs involved in exploiting this redundancy.

**Pruning and Compression** Many prior works focus on pruning weights [29, 13, 11, 15, 31, 9, 30] and how it can lead to sparse neural networks with many weights turned off. Our focus, however, is on understanding redundancy at the neuron level, without changing the weights. Work on structured pruning is more closely related to our work [33, 17], however, a key focus of these works is to prune channels/filters from convolution layers. Our work is more focused on understanding the nature of learned features and is more broadly applicable to all kinds of layers and models. We additionally focus on randomly pruning neurons, whereas structured pruning methods perform magnitude or feature-selection-based pruning.

**Explainability/Interpretability** Many works aim to understand learned representations with the goal of better explainability [35, 63, 2, 20, 40, 39, 10, 67]. Two works in this space are especially related to our work: sparse linear layers [61] which show that one can train sparse linear layers on top of extracted features from DNNs; and concept bottleneck models [21] which explicitly introduce a layer in which each neuron corresponds to a meaningful semantic concept. Both these works explicitly optimize for small/sparse layers, whereas our work shows that similar "small" layers already exist in pre-trained networks, and in fact, can be found simply with random sampling.

**Understanding Deep Learning** A related concept is that of instrinsic dimensionality of DNN landscapes [32]. Similar to our work, intrinsic dimensionality also requires dropping random parameters (weights) of the network. We, however, are concerned with dropping individual neurons. Other works on understanding deep learning [49, 1] have also looked at the learned features, however, none of these works analyze the redundancy at the neuron level. Another related phenomenon to diffused redundancy is that of neural collapse [42], which states that representations of the penultimate layer "collapse" to $K$ points (where $K =$ number of classes in the pre-training dataset). This implies that for perfectly collapsed representations we need to store just enough information in the final layer activations to be able to represent $K$ different points. We interestingly show that this information is spread throughout the layer with a significant degree of redundancy.

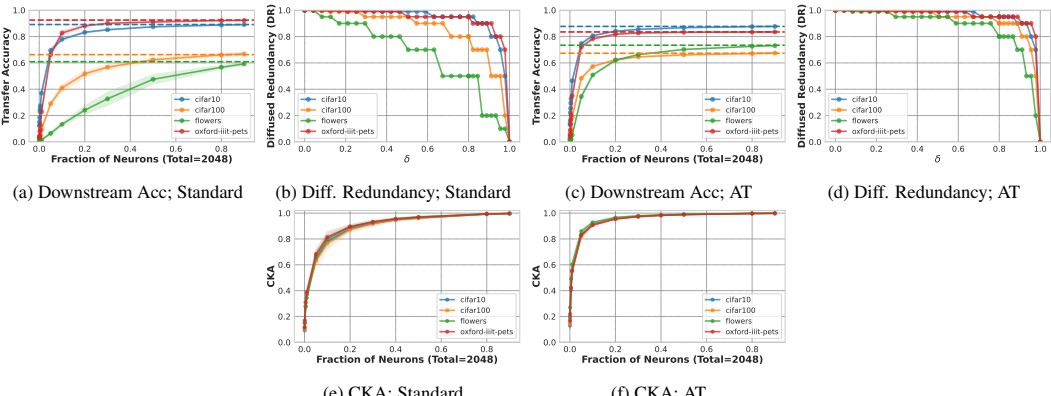

| (a) Downstream Acc; Standard | (b) Diff. Redundancy; Standard | (c) Downstream Acc; AT | (d) Diff. Redundancy; AT |
| --- | --- | --- | --- |

| (e) CKA; Standard | (f) CKA; AT |
| --- | --- |

Figure 1: **[Testing For Diffused Redundancy in ResNet50 Pre-trained on ImageNet1k]** Top: transfer accuracies + Diffused Redundancy ($DR$) measure (Eq 1) on different downstream datasets, dotted horizontal line shows accuracy obtained using the full layer. We see that accuracy obtained using parts of representation varies greatly with pre-training loss (much more diffused redundancy in Adversarially Trained (AT) ResNet), but also depends on the downstream dataset. Bottom: comparing CKA between a randomly chosen fraction of neurons to the whole layer. Here we evaluate CKA on samples from different datasets and find that similarity of a subset of layer rapidly increases, reaching a similarity of greater than $90\%$ on the adversarially trained ResNet with only $10\%$ randomly chosen neurons. Values are averaged over 5 random picks and error bars show std. dev.

## 2 The Diffused Redundancy Phenomenon

Prior observations about a *compression* phase [49] and neural collapse [42] suggest that the representations need not store a lot of information about the input. These findings imply that all neurons in a learned representation might not be necessary to capture all the information in a particular layer. Extending these observations, we propose the *diffused redundancy* hypothesis:

*Learned features are diffused throughout a given layer with redundancy such that there exist many randomly chosen subsets of neurons that can achieve similar performance to the whole layer for a variety of downstream tasks.*

Note that our hypothesis has two related but distinct parts to it: 1) redundancy in learned features, and 2) diffusion of this redundancy throughout the extracted feature vector. *Redundancy* refers to features being replicated in parts of the representation so that one can perform downstream tasks with parts of representation as well as with the full representation. *Diffusion* refers to this redundancy being spread all over the feature vector (as opposed to being structured), *i.e.*, *many* random subsets (of sufficient size) of the feature vector perform equally well.

In order to evaluate the *redundancy* part of the diffused redundancy hypothesis we use two tasks: 1) representation similarity between randomly chosen subsets of a representation with the whole representation, and 2) transfer accuracy on out-of-distribution datasets (using a linear probe) of randomly chosen subsets of the representation compared to the whole representation. To estimate *diffusion*, we run each check for redundancy over multiple random seeds and plot the standard deviation over these runs.

**Representation Similarity of Part vs Whole** Centered Kernel Alignment (CKA) is a widely used representation similarity measure and takes in two representations of $n$ data points $Z \in \mathbb{R}^{n \times d_1}$ and $Y \in \mathbb{R}^{n \times d_2}$ and gives a similarity score between 0 and 1 [24]. Intuitively, CKA (with linear kernel, see Appendix A for details about CKA) measures if the two representations rank the $n$ points similarly (where similarity is based on cosine distances). For a given neural network $g$ and $n$ samples drawn from a given data distribution, *i.e.*, $X \sim \mathcal{D}$, let $g(X)$ be the (penultimate) layer representation. If $m$ is a boolean vector representing a subset of neurons in $g(X)$, then we aim to measure $\text{CKA}(m \odot g(X), g(X))$ to estimate how much redundancy exists in the layer. If indeed $\text{CKA}(m \odot g(X), g(X))$ is high (*i.e.* close to 1) then it's a strong indication that the diffused redundancy hypothesis holds.

**Downstream Transfer Performance of Part vs Whole** A commonly used paradigm to measure the quality of learned representations is to measure their performance on a variety of downstream tasks [68, 22]. Here, we attach a linear layer ($h$) on top of the extracted features of a network ($g$) to

do classification. This layer is then trained using the training dataset of the particular task (keeping $g$ frozen). If features were to be diffused redundantly then accuracy obtained using $h \circ g$, *i.e.* linear layer attached to the entire feature vector, should be roughly the same as $h' \circ (m \odot g)$; where $m$ is a boolean vector representing a subset of neurons extracted by $g$, and $h$ & $h'$ are independently trained linear probes.

For both tasks, *i.e.* representation similarity and downstream transfer performance, we evaluate on CIFAR10/100 [25], Oxford-IIIT-Pets [43] and Flowers [38] datasets, from the VTAB benchmark [68]. Training and pre-processing details are included in Appendix B.

**Measure of Diffused Redundancy** In order to rigorously test our hypothesis, we define a measure of diffused redundancy $(DR)$ for a given model $(g)$ with $\mathcal{M}$ being a set of all possible boolean vectors of size $|g|$, *i.e.* size of the representation extracted from $g$. Each vector $m \in \mathcal{M}$ represents a possible subset of neurons from the entire layer. This measure is defined on a particular task $(T)$ as follows:

$$DR(g, T, \delta) = 1 - \frac{\min f \ , \text{s.t.} \ \frac{1}{|\mathcal{M}_f|} \sum_{m \in \mathcal{M}_f} \frac{T(m \odot g)}{T(g)} \geq \delta}{|g|}, \tag{1}$$

$$\mathcal{M}_f = \left\{ m \in \mathcal{M} | \sum_i m_i = f \ ; \ m \in \{0, 1\}^{|g|} \right\}$$

Here $T(.)$ denotes the performance of the model inside () for the particular task and $\delta$ is a user-defined tolerance level. For the task of representation similarity $T(m \odot g)$ is CKA between a subset of neurons denoted by $m \odot g$ and $g$, and $T(g)$ is always 1, since it denotes CKA between $g$ and $g$. For downstream transfer performance, $T(m \odot g)$ is the test accuracy obtained by training a linear probe on the portion of representation denoted by $m \odot g$ and $T(g)$ is the test accuracy obtained using the full representation. For $\delta = 1$, this measure tells what fraction of neurons could be discarded to exactly match the performance of the entire set of neurons. A higher value of $DR$ denotes that only a few random neurons were needed to match the task performance of the full set of neurons, and thus indicates higher redundancy. Since $\mathcal{M}$ contains an exponential number of vectors $(2^{|g|})$, precisely estimating this quantity is hard. Thus, we first choose a few $f$ (number of neurons to be chosen) to define subsets of $\mathcal{M}$. Then for each $\mathcal{M}_f$ we randomly select 5 samples.

## 2.1 Prevalence of Diffused Redundancy in Pre-Trained Models

Figure 1 checks for diffused redundancy in the penultimate layer representation of two types of ResNet50 pre-trained on ImageNet1k: one using the standard cross-entropy loss and another trained using adversarial training [34] (with $\ell_2$ threat model and $\epsilon = 3$) [3]. We check for diffused redundancy using both tasks of representation similarity and downstream transfer performance.

**Redundancy** This is indicated along the x-axis of Fig 1, *i.e.*, redundancy is shown when some small subset of the full set of neurons can achieve almost as good performance as the full set of neurons. When looking at downstream task performance (Figs 1a&1c), in order to obtain performance within some $\delta\%$ of the full layer accuracy (dotted lines), the fraction of neurons that can be discarded are task-dependent, *e.g.* across both training types we see that flowers (102 classes) and CIFAR100 (100 classes) require more fraction of neurons than CIFAR10 (10 classes) and oxford-iiit-pets (37 classes), perhaps because both these tasks have more classes. Additionally, across all datasets, the model trained with adversarial training exhibits more diffused redundancy than the one trained with standard loss (Fig 1d& 1b respectively), meaning we can discard far more neurons for the adversarially trained model to reach close to the full layer accuracy. Interestingly when looking at CKA between part of the feature vector with the full extracted vector (Figs 1e&1f), we do not see a significant difference in trends when evaluating CKA on samples from different datasets. However, we still see that we can achieve a given level of CKA with far fewer fraction of neurons in the adversarially trained ResNet50 as compared to the usually trained ResNet50.

*Diffused* **Redundancy** This is indicated by small error bars in Figs 1a&1c&1e&1f. If redundancy were instead very structured, then different random picks of neurons would have high variance,

---

[3]We also report results for $\ell_\infty, \epsilon = 4/255$ in Appendix D and generally find that the $\ell_2$ models exhibit higher diffused redundancy. We choose to study adversarially robust models since they've been shown to transfer better [46].

however, the error bars here are very low, showing that performance is very stable across different random picks, thus indicating that redundancy is diffused throughout the layer.

While both tasks of downstream transfer and CKA between part and whole indicate higher diffused redundancy for the adversarially trained model, we see that downstream transfer performance can differ substantially based on the dataset (while CKA remains fairly stable across the same datasets), indicating that downstream performance turns out to be a "harder" test for diffused redundancy. Thus, in the rest of the paper, we examine diffused redundancy through the lens of downstream transfer performance and include CKA results in Appendix A.

## 2.2 Understanding Why *Many* Random Subsets Work

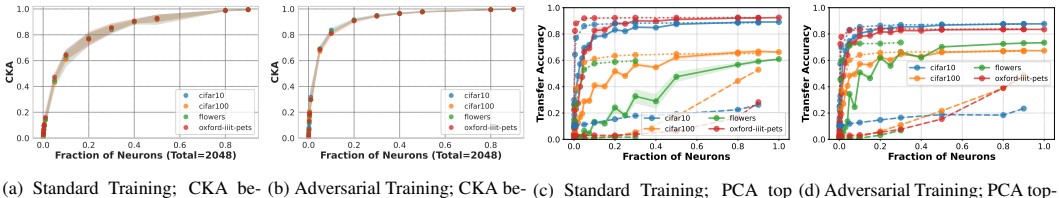

(a) Standard Training; CKA between two random subsets
(b) Adversarial Training; CKA between two random subsets
(c) Standard Training; PCA top (dotted) and bottom (dashed)
(d) Adversarial Training; PCA top-k (dotted) and bottom-k (dashed)

Figure 2: **[Why Many Random Subsets Work]** (2a & 2b) Similarity between any two randomly picked sets of $k\%$ neurons becomes fairly high (for a "critical mass" of $k\%$), thus showing that any random pick beyond this threshold is likely to perform similarly; (2c & 2d) Performance of a linear probe trained for a downstream task on randomly chosen neurons (solid lines) closely follows that of a linear probe trained on a projection on top $k$ principal components (dotted lines) for a sufficiently large value of $k$.

Many prior works explicitly train models to have "small" representations (*e.g.* [26, 65, 64, 5] with the goal of efficient downstream learning. These works show that, when explicitly optimized, networks can perform downstream classification with fewer neurons than typically used in state-of-the-art architectures. We show, however, that such subsets already exist in models that are *not* explicitly trained for this goal, and in fact, one doesn't even have to try hard to find this subset; it can be *randomly* chosen. Later in section 3.3 we compare some of these efficient representation learning methods to randomly chosen subsets and carefully analyze the tradeoffs involved. Here, however, we seek to better understand why there exist so many randomly selected subsets that work just as well as the whole layer.

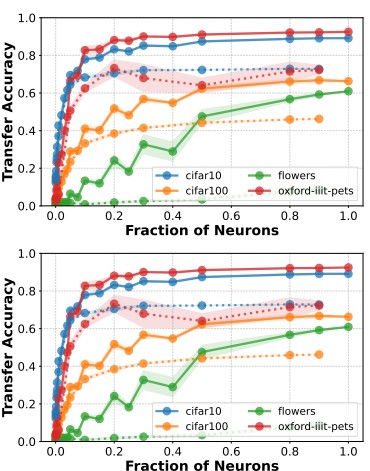

Figure 3: **[Random Projection (dotted) vs Randomly Choosing Neurons (solid)]** We find that diffused redundancy (randomly choosing neurons) performs significantly better than random projections. This indicates that the "constrained" projection offered by diffused redundancy is crucial in achieving good downstream performance.

**High CKA between two random subsets of the same size.** First, we use representation similarity (CKA) [24] to get a sense of how two random picks of neurons (of the same size) relate to each other. Concretely, we calculate CKA between two random picks of $k\%$ neurons in the penultimate layer (averaged over 10 such randomly picked pairs) on samples taken from different datasets. Fig 2a & 2b show CKA results averaged over these different picks of pairs of subsets of the full set of neurons. We see that after picking a certain threshold, *i.e.* for a large enough value of $k$, the similarity between any two randomly picked pairs of heads is fairly high. For example, for the adversarially trained ResNet50 (Fig 2b), we observe that any $10\%$ of neurons picked from the penultimate layer are highly similar (CKA of about $0.8$), with very low error bars. A similar value of CKA is obtained with $20\%$ of neurons for the standard ResNet50 model. These results indicate that, given a sufficient size, picking any random subset of that size has very similar representations and thus provides an initial intuition for why almost *any* random subset, with high probability, could work equally well.

**Downstream performance on $k$ randomly chosen neurons closely follows performance on top $k$ principal components.** While high representation similarity is a necessary condition for two representations to have similar downstream performance, it's not a sufficient one. As seen earlier in Fig 1, similar values of CKA can still show different downstream accuracy. Thus, to further understand why any random pick of neurons achieves similar accuracy, we compare a randomly chosen subset of neurons with a projection on the same number of top $k$ principal components. Fig 2c& 2d shows that the performance of a linear probe trained on a random subset of neurons initially lags behind the top PCA dimensions for small values of $k(< 20\%)$, but for higher values $(> 20\%)$ closely follows the performance on a probe trained on the projection on the same number of top principal components. This indicates that a sufficiently sized random subset of size $k$ (roughly) captures the maximum variance in data that can be captured in $k$ dimensions (upper bound by top $k$ principal components since they are designed to capture maximum variance). As a sanity check, we also show the results for the bottom $k$ principal components (directions of least variance) and see that randomly chosen neurons perform significantly better.

**Random projections into lower dimensions do not perform well on downstream tasks.** The presence of diffused redundancy in a particular layer indicates that a dimension lower than the layer's size suffices to perform many downstream tasks. However, randomly sampling neurons (as we do in our experiments on diffused redundancy) can be seen as one very particular way of reducing dimension that is restricted by what the network has learned. This raises a natural question, would our observation extend to less restricted ways of reducing dimensions? To test this, we compare diffused redundancy *i.e.*, a linear probe trained on a random sample of $k$ neurons to a linear probe trained on a random projection of the layer's activations. To project a $d$ dimensional layer into a lower dimension $(k)$ we multiply it by a (normalized) randomly sampled matrix from a normal distribution $\mathbb{R}^{d \times k}$. We report our results over 5 seeds in Fig 3 and find that diffused redundancy significantly outperforms random projection.

## 3 Factors Influencing The Degree of Diffused Redundancy

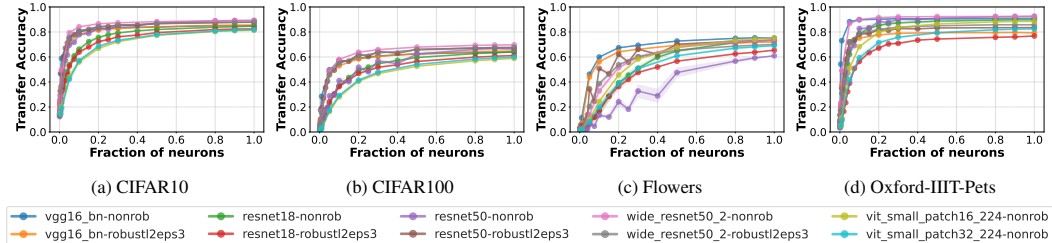

|              |              |              |              |
|--------------|--------------|--------------|--------------|
| (a) CIFAR10  | (b) CIFAR100 | (c) Flowers  | (d) Oxford-IIIT-Pets |

- vgg16_bn-nonrob
- vgg16_bn-robustl2eps3
- resnet18-nonrob
- resnet18-robustl2eps3
- resnet50-nonrob
- resnet50-robustl2eps3
- wide_resnet50_2-nonrob
- wide_resnet50_2-robustl2eps3
- vit_small_patch16_224-nonrob
- vit_small_patch32_224-nonrob

Figure 4: **[Comparisons Across Architectures For Downstream Task Accuracy]** All models shown here are pre-trained on ImageNet1k. We see that diffused redundancy exists across architectures, and the trend observed in Figure 1c&1a regarding adversarially trained models also holds here as models' curves that are more "inside" are the ones trained with standard loss.

In order to better understand the phenomenon of diffused redundancy we analyze 30 different pre-trained models, with different architectures, pre-training datasets and losses. We then evaluate each model for transfer accuracy on 4 datasets mentioned in Section 2. While all results in this section are on the penultimate layer, we also report results on intermediate layers in Appendix E and find similar trends of diffused redundancy. All details for reproducibility can be found in Appendix B.

**Architectures** We consider VGG16 [50], ResNet18, ResNet50, WideResNet50-2 [16], ViT-S16 & ViT-S32 [23]. Additionally, we consider ResNet50 with varying widths of the final layer (denoted by `ResNet50_ffx` where x denotes the number of neurons in the final layer).

**Upstream Datasets** ImageNet-1k & ImageNet-21k [45].

**Upstream Losses** Standard cross-entropy, adversarial training ($\ell_2$ threat model, $\epsilon = 3$ and $\ell_\infty$ threat model, $\epsilon = 4/255$ [4]) [34], MRL Loss [26], DeCov [6], and varying strengths of dropout [52].

**Downstream Datasets** CIFAR10/1000, Oxford-IIIT-Pets, and Flowers, same as Section 2. Additionally, we also report performance on harder datasets such as ImageNetv2 [44] and Places365 [69]

---

[4]Results for $\ell_\infty$ robust models can be found in Appendix D

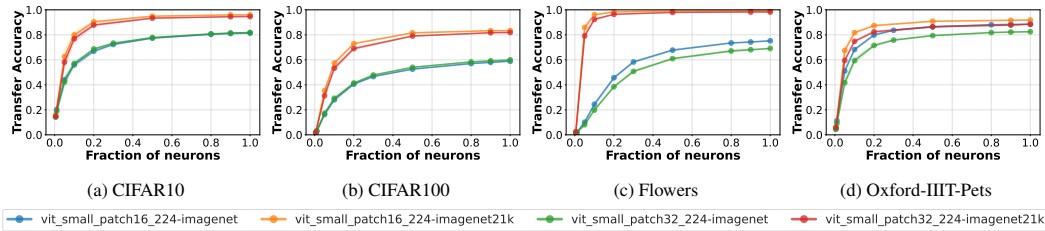

| (a) CIFAR10 | (b) CIFAR100 | (c) Flowers | (d) Oxford-IIIT-Pets |

vit_small_patch16_224-imagenet    vit_small_patch16_224-imagenet21k    vit_small_patch32_224-imagenet    vit_small_patch32_224-imagenet21k

Figure 5: **[Comparison Across Upstream Datasets]** We see that degree of diffused redundancy depends a great deal on the upstream training dataset, in particular, models trained on ImageNet21k exhibit a higher degree of diffused redundancy, although the differences in the degree of diffused redundancy are downstream task dependent

in Appendix F. For all analyses in this section, we also report corresponding approximations for $DR$ (Eq 1) in Appendix D.

## 3.1 Effects of Architecture, Upstream Loss, Upstream Datasets, and Downstream Datasets

Extending the analysis in Section 2, we evaluate the diffused redundancy hypothesis on other archi-tectures. Fig 4 shows transfer performance for different architectures. All architectures shown in Fig 4 are trained on ImageNet1k. We find that our takeaways from Section 2 also extend to other architectures.

Fig 5 compares two instances each of ViT-S16 and ViT-S32, one trained on a bigger upstream dataset (ImageNet21k) and another on a smaller dataset (ImageNet1k)

Note that the nature of all curves in both Figs 4&5 highly depends on downstream datasets. This is also consistent with the initial observation of Section 2 about diffused redundancy being downstream dataset dependent. Additionally, we also report results on ImageNetV2 and Places365 in Appendix F showing that diffused redundancy also holds on "harder" datasets.

## 3.2 Diffused Redundancy as a Function of Layer Width

We take the usual ResNet50 with a penultimate layer consisting of 2048 neurons and compare it with variants that are pre-trained with a much smaller penultimate layer, these are denoted by `ResNet50_ffx` where x ($< 2048$) is the number of neurons in the penultimate layer. Fig 6 shows how diffused redundancy slowly fades away as we squeeze the layer to be smaller. In fact, for `ResNet50_ff8`, we see that across all datasets we need $> 90\%$ of the full layer to achieve perfor-mance close to the full layer. This shows that diffused redundancy only appears in DNNs when the layer is sufficiently wide to encode redundancy.

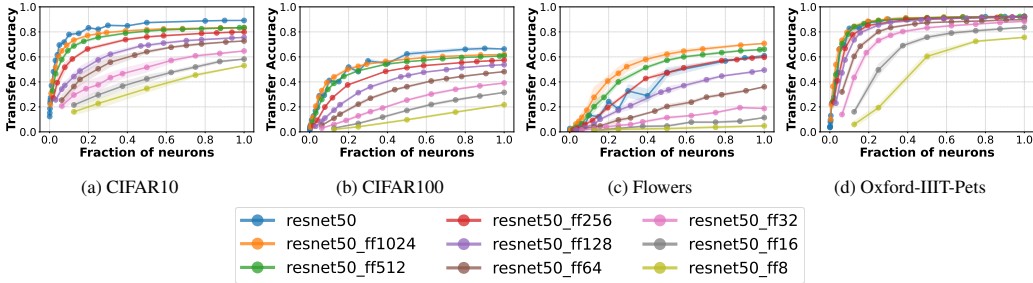

| (a) CIFAR10 | (b) CIFAR100 | (c) Flowers | (d) Oxford-IIIT-Pets |

| resnet50 | resnet50_ff256 | resnet50_ff32 |
| resnet50_ff1024 | resnet50_ff128 | resnet50_ff16 |
| resnet50_ff512 | resnet50_ff64 | resnet50_ff8 |

Figure 6: **[Diffused Redundancy as Function of Layer Width]** As we make the length of the layer smaller, the degree of redundancy becomes lesser. For `ResNet50_ff8`, *i.e.* ResNet50 with only 8 neurons in the final layer, we see that we need almost $90\%$ of neurons to achieve similar accuracy as the full layer.

## 3.3 Comparison With Methods That Optimize For Lesser Neurons

Matryoshka Representation Learning (MRL) is a recently proposed paradigm that learns nested representations such that the first $k, 2k, 4k, ..., N$ (where $N$ = size of the full layer) dimensions of the extracted feature vector are all explicitly made to be good at minimizing upstream loss, with

the intuition of learning coarse-to-fine representations. This ensures that one can flexibly use these smaller parts of the representation for downstream tasks. MRL, thus, ensures that redundancy shows up in learned representations in a *structured* way, *i.e.*, we know the first $k, 2k, ...$ neurons can be picked and used for downstream tasks and should perform reasonably.

Here we investigate two questions regarding Matryoshka representations: 1) do these representations also exhibit the phenomenon of diffused redundancy? *i.e.* if we were to ignore the structure imposed by MRL-type training and instead just pick random neurons from all over the layer, do we still get reasonable performance?, and 2) how do they compare to representations learned by other kinds of losses?

Figure 7 investigates these questions by comparing ResNet50 representations learned using MRL loss to other losses. `resnet50_mrl_nonrob_first` (red line) denotes a ResNet50 trained using MRL loss and evaluated on parts of the representation that were optimized to have low upstream loss (*i.e.* first $k, 2k, ...N$ neurons, here $k = 8$ and $N = 2048$) and `resnet50_mrl_nonrob_random` (green line) refers to the same model with the same number of neurons chosen for evaluation, except they're chosen at random from the entire layer.

First, we interestingly see that even the ResNet50 trained with MRL loss exhibits diffused redundancy (denoted by green line spiking very quickly for most datasets in Fig 7), despite having been trained to only have structured redundancy. Based on this observation, we conjecture that diffused redundancy is a natural consequence of having a wide layer. Second, we see that ResNet50 trained on MRL indeed does better in the low neuron regime across datasets (red line on the extreme left part of the plots in Fig 7), but other models quickly catch up as we pick more neurons, thus indicating that major efficiency benefits of MRL-type models are best realized when using an extremely low number of neurons, else one can obtain similar downstream performances by simply picking random samples from existing pre-trained models.

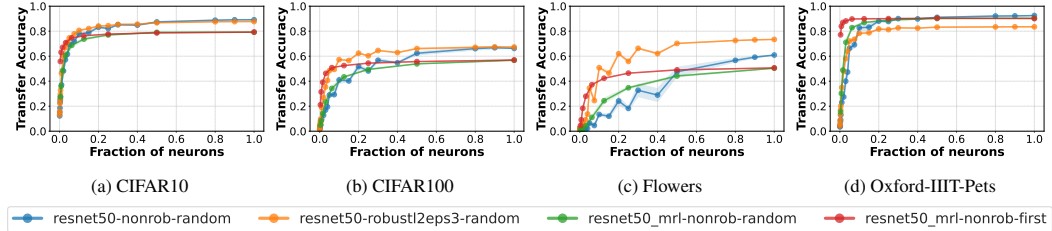

Figure 7: **[Comparison of Diffused Redundancy in MRL vs other losses]** Here we compare ResNet50 trained using multiple losses including MRL [26]. `nonrob` indicates that the model was trained with the standard crossentropy loss while `robustl2eps3` indicates adversarial training with $\ell_2$ threat model and $\epsilon = 3$. Red line shows results for part of the representation explicitly optimized in MRL, whereas the green line shows results for parts that are picked randomly from the same representation. Even the MRL model shows a significant amount of diffused redundancy despite being explicitly trained to instead have structured redundancy.

### 3.4 Methods That Prevent Co-adaptation of Neurons Also Exhibit Diffused Redundancy

We consider two additional modifications to upstream pre-training: we add regularization to the usual cross-entropy loss that decorrelates neurons in the feature representation using DeCov [6] and Dropout [52]. Dropout does this implicitly by randomly dropping neurons during training and DeCov does this explicitly by putting a loss on the activations of a given layer. We pre-train different ResNet50 on ImageNet1k by varying strengths of dropout ranging from 0.1 all the way up to 0.8 and also by separately adding the DeCov regularizer (regularization strength $= 1e-4$) to give 9 additional pre-trained models. Intuitively these methods should lead to *increased* diffused redundancy since by design these methods force the model to disperse similar information in different parts of a layer to perform the same task downstream task. Interestingly, we see that diffused redundancy in these models is almost completely independent of the regularization strength. This suggests that diffused redundancy might be an inevitable property of DNNs when the layer has sufficient width. Due to space constraints, we include these results in Appendix G.

## 4 Possible Fairness-Efficiency Tradeoffs in Efficient Downstream Transfer

One natural use case for diffused redundancy is efficient transfer to downstream datasets. As defined in Eq 1 and as also seen in Sections 2 & 3, dropping neurons comes at a small cost ($\delta$ in Eq 1) in

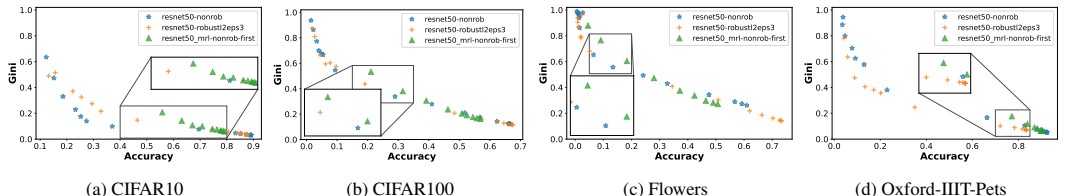

|  (a) CIFAR10 | (b) CIFAR100 | (c) Flowers | (d) Oxford-IIIT-Pets |

Figure 8: **[Gini Coefficient of Class-Wise Accuracies as we Drop Neurons]** Higher value of Gini coefficient indicates higher inequality [12]. We see that for all models gini coefficients become higher as the accuracy reduces (as a result of dropping neurons). Additionally, in some regions (highlighted in the plots), the model explicitly optimized for efficient transfer (`resnet50_mrl`) can give rise to higher gini values, resulting in a more unequal spread of accuracy over classes.

performance as compared to the full set of neurons. Here we take a deeper look into this drop in overall performance and investigate how it is distributed across classes. If the drop affects only a few classes, then dropping neurons – although efficient for downstream tasks – could have implications for fairness, which is not only of concern to ML researchers and practitioners [66, 14, 18], but also to lawyers [56] and policymakers [57].

We compare the spread of accuracies across classes using inequality indices, which are commonly used in economics to study income inequality [8, 47] and have also recently been adopted in the fair ML literature [51]. We use gini index [12] and coefficient of variation [27] to quantify the spread of performance across classes. For a perfect spread, both gini and coefficient of variation are 0, and higher values indicate higher inequality.

Figure 8 compares the gini index for various models at varying levels of accuracy (note that accuracy monotonically increases with more neurons, hence the right most point for a model represents the model with all neurons). We make two observations: across all datasets and all models we find that a loss in accuracy (compared to the full layer) comes at the cost of a few classes, as opposed to being smeared throughout classes, as indicated by high gini values on the left of each plot. Additionally, we observe that the model trained using MRL loss tends to have slightly higher gini values in the regions where the drop in accuracy is slightly higher (highlighted on the plots). To ensure that this trend is not simply due to lower accuracy, we investigate the error distributions across classes (Appendix C) and find that predictions become more homogeneous as we drop more neurons. Similar trends are also observed with coeff. of variation as shown in Appendix C. These results draw caution to the potential unintended side-effects of exploiting diffused redundancy and suggest that there could be a possible fairness-efficiency tradeoff involved.

**Connections to Robustness-Fairness Tradeoffs** There are well-established tradeoffs between robustness and fairness in both standard [36] and adversarial training [62, 58]. One major difference between these works and our setup is that the task they train on is also the task they test on. We instead operate in the transfer learning setup where we train a linear probe on the representation of a pre-trained model. While it's not immediately clear whether the discrepancy between class accuracies observed on the pre-training task (*e.g.* as in [62]) also leads to an inter-class accuracy discrepancy on downstream tasks when dropping neurons, it would be very interesting future work to establish more formal connections between the robustness-fairness tradeoff and its effects on the fairness-efficiency tradeoff presented in this paper.

## 5   Conclusion and Broader Impacts

We introduce the diffused redundancy hypothesis and analyze a wide range of models with different upstream training datasets, losses, and architectures. We carefully analyze the causes of such redundancy and find that upstream training (both loss and datasets) plays a crucial role and that this redundancy also depends on the downstream dataset. One direct practical consequence of our observation is increased efficiency for downstream training times which can have many positive impacts in terms of reduced energy costs [54] which is crucial in moving towards "green" AI [48]. We, however, also draw caution to potential pitfalls of such efficiency gains , which might hurt the accuracy of certain classes more than others, thus having direct consequences for fairness.

## Acknowledgements

VN and JPD were supported in part by NSF CAREER Award IIS-1846237, NSF D-ISN Award #2039862, NSF Award CCF-1852352, NIH R01 Award NLM013039-01, NIST MSE Award #20126334, DARPA GARD #HR00112020007, DoD WHS Award #HQ003420F0035, ARPA-E Award #4334192 and a Google Faculty Research Award. VN, TS, and KPG were supported in part by an ERC Advanced Grant "Foundations for Fair Social Computing" (no. 789373). AW acknowledges support from a Turing AI Fellowship under grant EP/V025279/1 and the Leverhulme Trust via CFI. SF was supported in part by a grant from an NSF CAREER AWARD 1942230, ONR YIP award N00014-22-1-2271, ARO's Early Career Program Award 310902-00001, Meta grant 23010098, HR00112090132 (DARPA/RED), HR001119S0026 (DARPA/GARD), Army Grant No. W911NF2120076, the NSF award CCF2212458, an Amazon Research Award, and an award from Capital One.

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

# A   Measuring Diffused Redundancy

## A.1   CKA Definition

In all our evaluations we use CKA with a linear kernel [24] which essentially amounts to the following steps:

1. Take two representations $Y \in \mathbb{R}^{n \times d1}$ and $Z \in \mathbb{R}^{n \times d2}$

2. Compute dot product similarity within these representation, *i.e.* compute $K = YY^T$, $L = ZZ^T$

3. Normalize $K$ and $L$ to get $K' = HKH$, $L' = HLH$ where $H = I_n - \frac{1}{n}\mathbf{1}\mathbf{1}^T$

4. Return $\text{CKA}(Y, Z) = \frac{\text{HSIC}(K,L)}{\sqrt{\text{HSIC}(K,K)\text{HSIC}(L,L)}}$, where $\text{HSIC}(K, L) = \frac{1}{(n-1)^2}(\text{flatten}(K') \cdot \text{flatten}(L'))$

We use the publicly available implementation of [37], which provides an implementation that can be calcuated over multiple mini-batches: `https://github.com/nvedant07/STIR`

## A.2   Additional CKA results

Fig 9 shows CKA comparison between randomly chosen parts of the layer and the full layer for different kinds of ResNet50. We observe that even ResNet50 trained with MRL loss shows a significant amount of diffused redundancy.

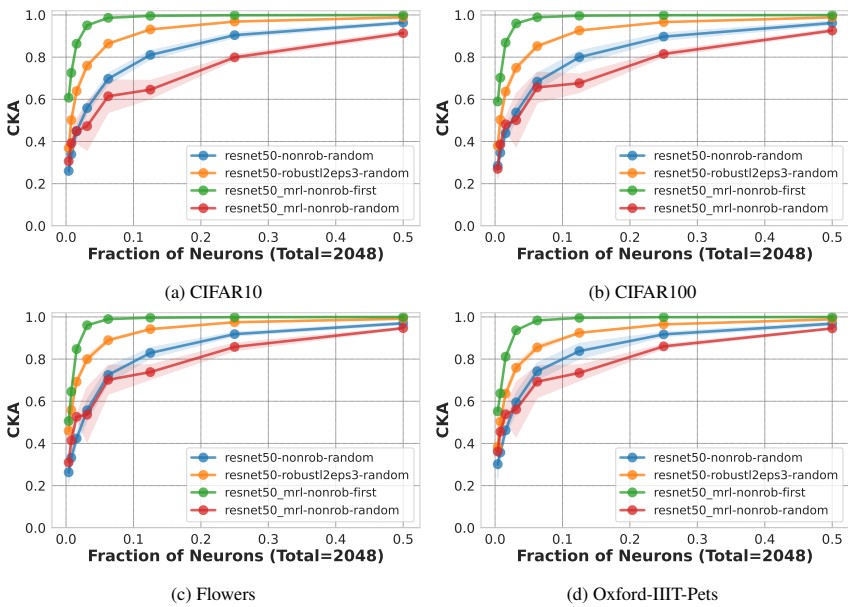

Figure 9: **[Comparison of Diffused Redundancy in MRL vs other losses, through the lens of CKA]** We see a similar trend as reported in Fig 7 in the main paper, where even the MRL model shows a significant amount of diffused redundancy despite being explicitly trained to instead have structured redundancy. The amount of diffused redundancy however is much lesser than the resnets trained using the standard loss and adv. training as denoted by a much lower red line across all datasets.

# B   Training and Pre-Processing Details for Reproducibility

Here we list the sources of weights for the various pre-trained models used in our experiments:

- ResNet18 trained on ImageNet1k using standard loss: taken from `timm` v0.6.1.
- ResNet18 trained on ImageNet1k with adv training: taken from Salman et al. [46]:

- ResNet50 trained on ImageNet1k using standard loss: taken from `timm` v0.6.1.
- ResNet50 trained on ImageNet1k with adv training: taken from Salman et al. [46]: `https://github.com/microsoft/robust-models-transfer`.
- ResNet50 trained on ImageNet1k using MRL and with different final layer widths (`resnet50_ffx`): taken from released weights of by Kusupati et al. [26]: `https://github.com/RAIVNLab/MRL`.
- WideResNet50-2 on ImageNet1k both standard and avd. training: taken from Salman et al. [46]: `https://github.com/microsoft/robust-models-transfer`.
- VGG16 trained on ImageNet1k with standard loss: taken from `timm` v0.6.1.
- VGG16 trained on ImageNet1k with adv training: taken from Salman et al. [46]: `https://github.com/microsoft/robust-models-transfer`.
- ViTS32 & ViTS16 trained on ImageNet21k & ImageNet1k: taken from weights released by Steiner et al. [53]: `https://github.com/google-research/vision_transformer`.

All linear probes trained on the representations of these models are trained using SGD with a learning rate of $0.1$, momentum of $0.9$, batch size of $256$, weight decay of $1e-4$. The probe is trained for $50$ epochs with a learning rate scheduler that decays the learning rate by $0.1$ every $10$ epochs. Scripts for training can also be found in the attached code.

For pre-processing, we re-size all inputs to 224x224 (size used for pre-training) and apply the usual composition of RandomHorizontalFlip, ColorJitter(brightness=0.25, contrast=0.25, saturation=0.25, hue=0.25), RandomRotation(degrees=2). All inputs were mean normalized. For imagenet1k pre-trained models: mean = [0.485, 0.456, 0.406] and std = [0.229, 0.224, 0.225]. For imagenet21k pre-trained models: mean = [0.5,0.5,0.5], std = [0.5,0.5,0.5].

## C  Deeper Analysis of Fairness-Efficiency Tradeoff in Section 4

**Analyzing Error Distributions** To ensure that the higher gini coefficient shown in Fig 8 as we drop more neurons is not merely an artifact of lower overall accuracy, we plot class-wise accuracies as we drop neurons (Figs 11, 12 & 13). We find that for the entire layer, accuracy starts at an almost uniform distribution, and while overall accuracy deteriorates as we drop neurons, the drop comes at a larger cost for a few classes resulting in disparate inter-class accuracies.

**Coeff of Variation for Measuring Inequality in Inter-Class Accuracy** Fig 10 shows results for the same analysis shown in Fig 8 of the main paper and we find similar takeaways even when using the coefficient of variation as a measure of inequality.

## D  Corresponding Diffused Redundancy Estimates For Analyses in Section 3 & $\ell_\infty$ Robust Model Results

Corresponding diffused redundancy (DR) ablations for Figures 4,5,7. These are shown in Figures 15,16,17 respectively. This should allow for easy comparison of diffused redundancy (lines that are more outside have higher DR). For example, Figure 16 clearly shows higher diffused redundancy in models trained on larger upstream datasets (here ImageNet21k) since these curves lie more on the outside of the same model's curves for ImageNet1k.

Additionally, we show numbers on x-axis for Figure 4 in Figure 18. Figures 5 and 7 compare models with same number of neurons in the final layer and hence trends shown with fraction on the x-axis will be exactly the same with absolute numbers on the x-axis. However, Figure 18 allows a direct comparison of the performance of the same absolute number of neurons across different models.

We report results for $\ell_\infty$ robust models (with $\epsilon = 4/255$) in Fig 14 and find that $\ell_2$ model generally shows a greater degree of diffused redundancy.

## E  Results on Intermediate Layers

We additionally ran our experiment on other intermediate layers and report the results in Fig 19. We present results for a ResNet50 pretrained on ImageNet1k using the standard CrossEntropy loss. The

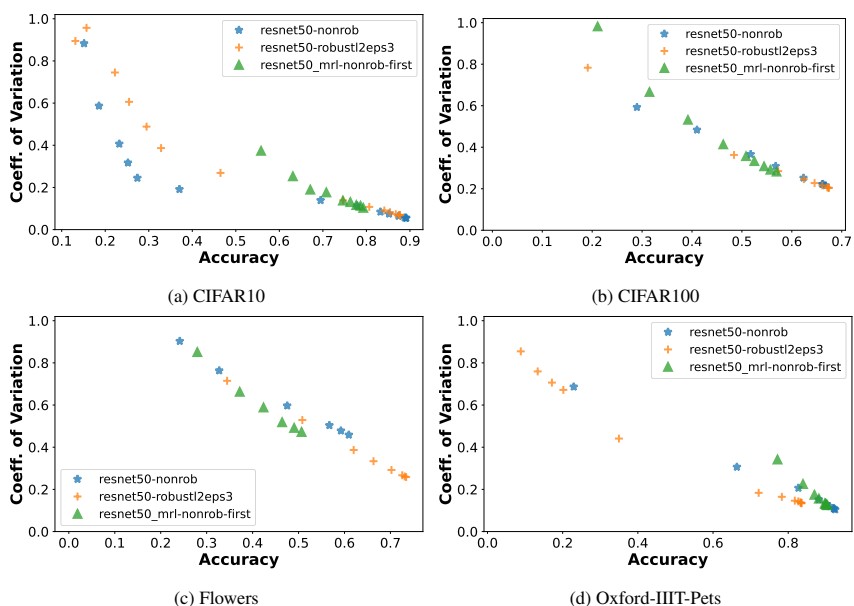

Figure 10: **[Coefficient of Variation As We Drop Neurons]** We see a similar trend as reported in Fig 8 of the main paper where inequality increases as we drop neurons for all models on all datasets.

intermediate layers considered are characterized as activations following each residual connection within distinct ResNet blocks. `layerX.Y.act3` means the $Yth$ residual connection in the $Xth$ ResNet block and act3 indicates that we're taking the value after the activation (ReLU) has been applied.

# F   Results on Harder Downstream Tasks: ImageNetV2 and Places365

We report results on harder downstream tasks such as ImageNet1k, ImageNetV2, and Places365 in Figure 20. We find that when randomly dropping neurons, the model is still able to generalize to ImageNet1k and Places365 with very few neurons, *i.e.*, the phenomena of diffused redundancy observed for smaller datasets, also holds for harder datasets. Interestingly we also observe that the accuracy gap between ImageNet1k and ImageNetV2 is maintained even as we drop neurons.

# G   Effects of Explicitly Preventing Co-adaptation of Neurons: Analysis of Dropout and DeCov

Regularizers such as dropout and DeCov, force different parts of the representations to not be correlated. Thus these regularizers can be seen as explicitly requiring different, compact parts of the representation to be self-contained for the downstream task. Thus, intuitively, such methods should increase diffused redundancy. Here we investigate if our observation about diffused redundancy is influenced by such regularizers. We evaluate ResNet50 pre-trained on ImageNet1k with dropout in the penultimate layer ranging from a strength of 0.1 all the way to 0.8. We also train another ResNet50 model with the DeCov regularizer added to the usual crossentropy loss and put a weight of 0.0001 on the regularizer to ensure that its numerical range is similar to that of the cross entropy loss term.

Results in Figures 21 & 22 suggest that such regularizers have *almost no effect* on diffused redundancy. Trends across datasets remain consistent regardless of the strength of dropout or the weight given to DeCov regularizer. This observation further adds to the evidence that diffused redundancy is likely to be a natural property of representations learned by DNNs.

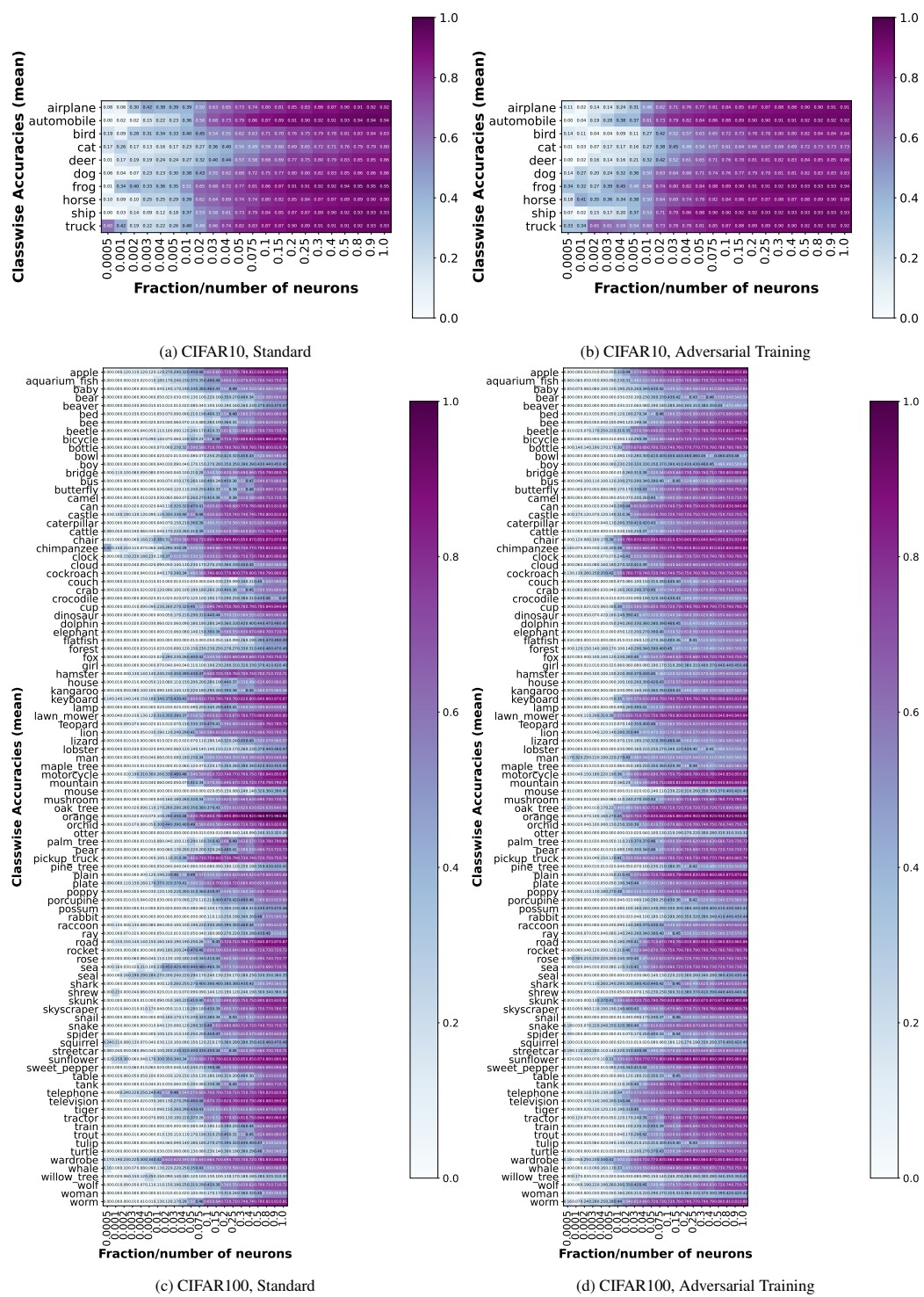

Figure 11: **[Error Distributions As A Function of Fraction of Neurons]** We see that accuracy deteriorates as we drop neurons, however, this drop comes at a larger cost for a few classes and results in near homogenous predictions for the least number of neurons on the left.

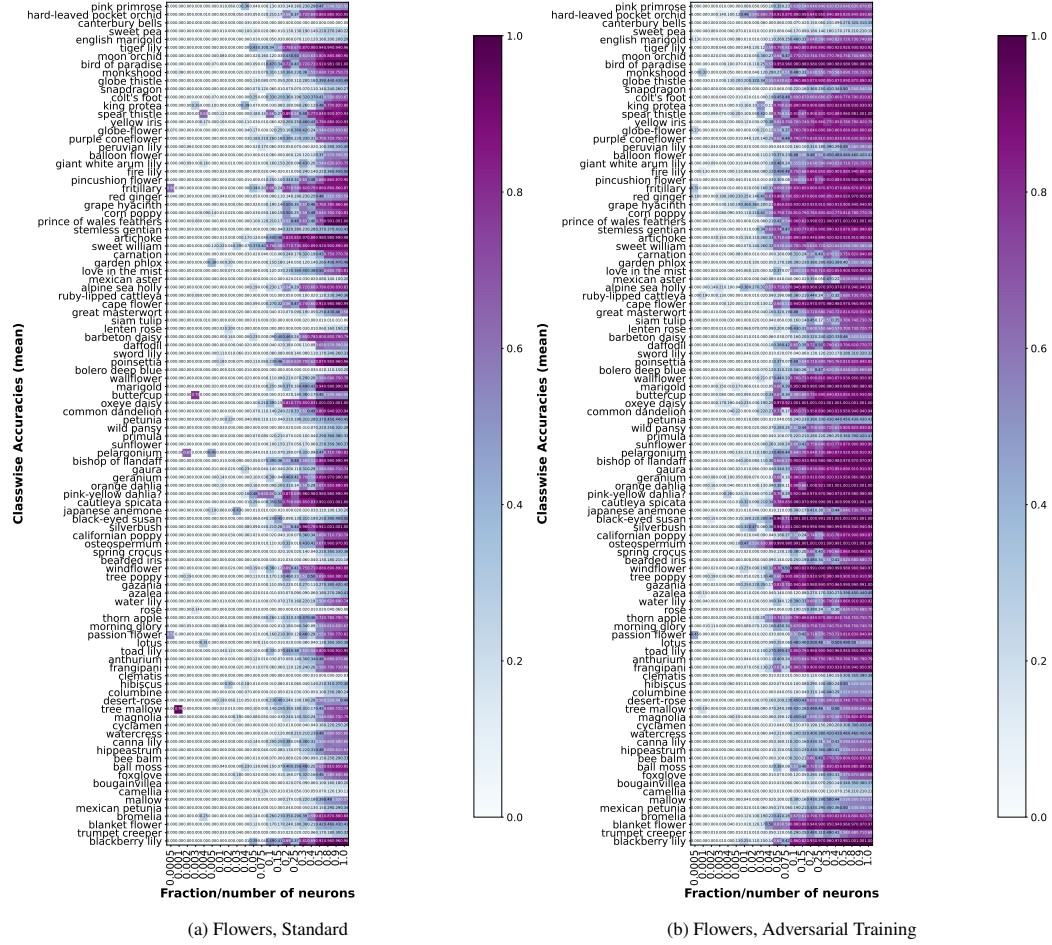

(a) Flowers, Standard

(b) Flowers, Adversarial Training

Figure 12: **[Error Distributions As A Function of Fraction of Neurons – continued]** We see that accuracy deteriorates as we drop neurons, however, this drop comes at a larger cost for a few classes and results in near homogenous predictions for the least number of neurons on the left.

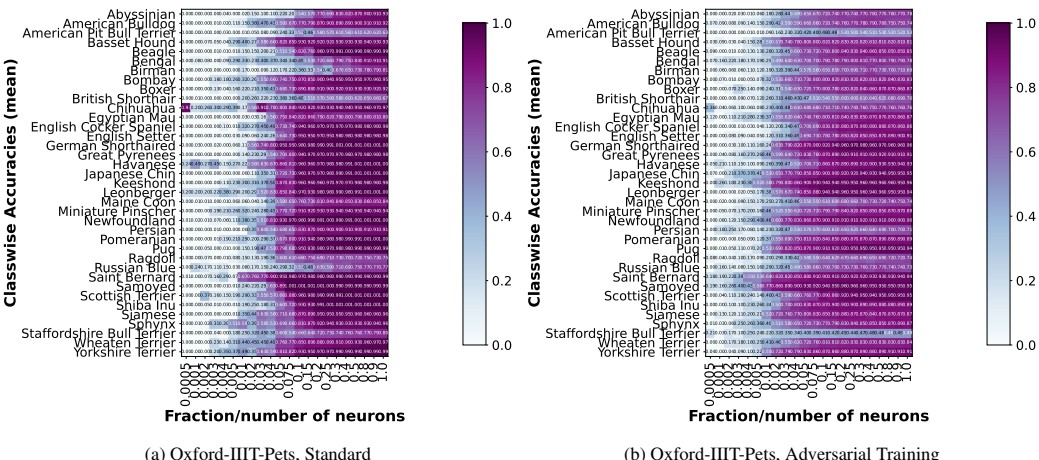

(a) Oxford-IIIT-Pets, Standard

(b) Oxford-IIIT-Pets, Adversarial Training

Figure 13: **[Error Distributions As A Function of Fraction of Neurons – continued]** We see that accuracy deteriorates as we drop neurons, however, this drop comes at a larger cost for a few classes and results in near homogenous predictions for the least number of neurons on the left.

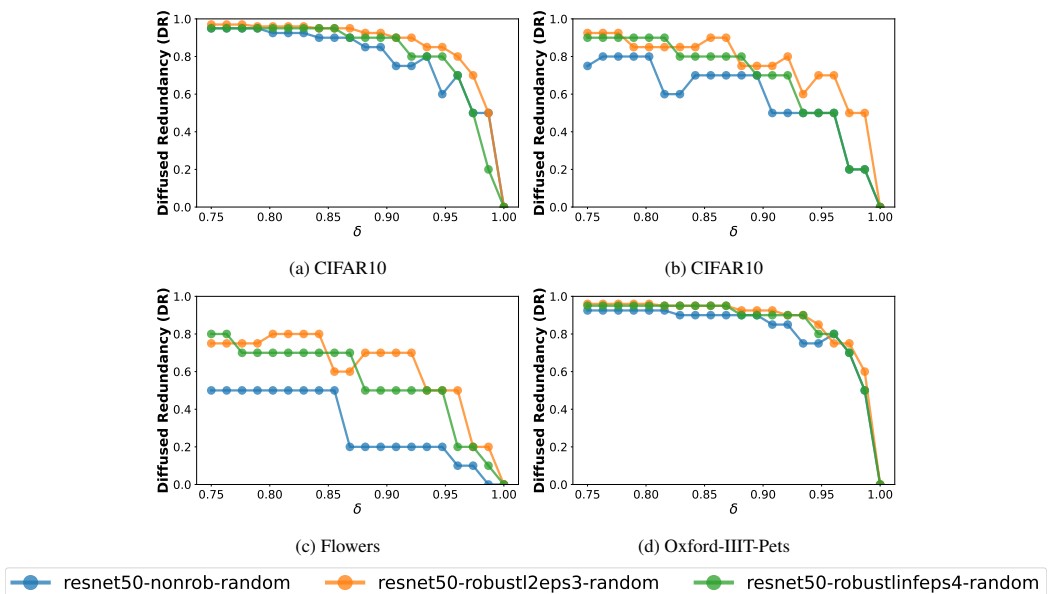

Figure 14: **[Results for $\ell_\infty$ robust model]** We show results for ResNet50 trained with 3 different losses: CrossEntropy (nonrob), Adversarial Training with $\ell_2$ threat model (robl2eps3), and with the $\ell_\infty$ threat model (roblinfeps4). We see that the $\ell_2$ threat model shows the most diffused redundancy. All models are trained on ImageNet1k.

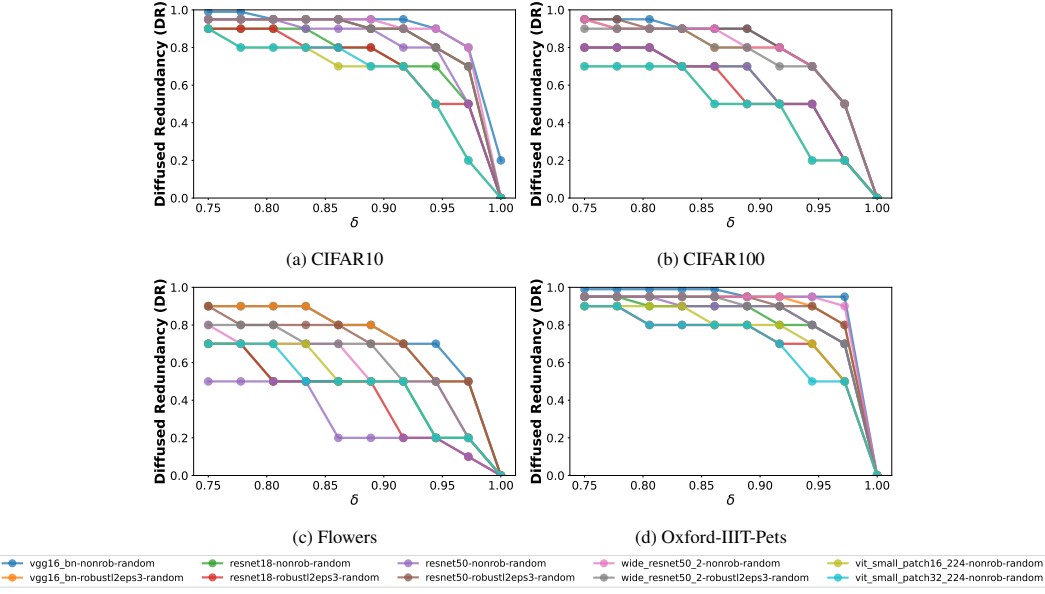

Figure 15: **[Comparisons Across Architectures For Downstream Task Accuracy]** All models shown here are pre-trained on ImageNet1k. This Figure shows corresponding diffused redundancy values for Figure 4 different $\delta$ values. We see that diffused redundancy exists across architectures, and the trend observed in Figure 1c&1a regarding adversarially trained models also holds here as models curves that are more "inside" are the ones trained with standard loss.

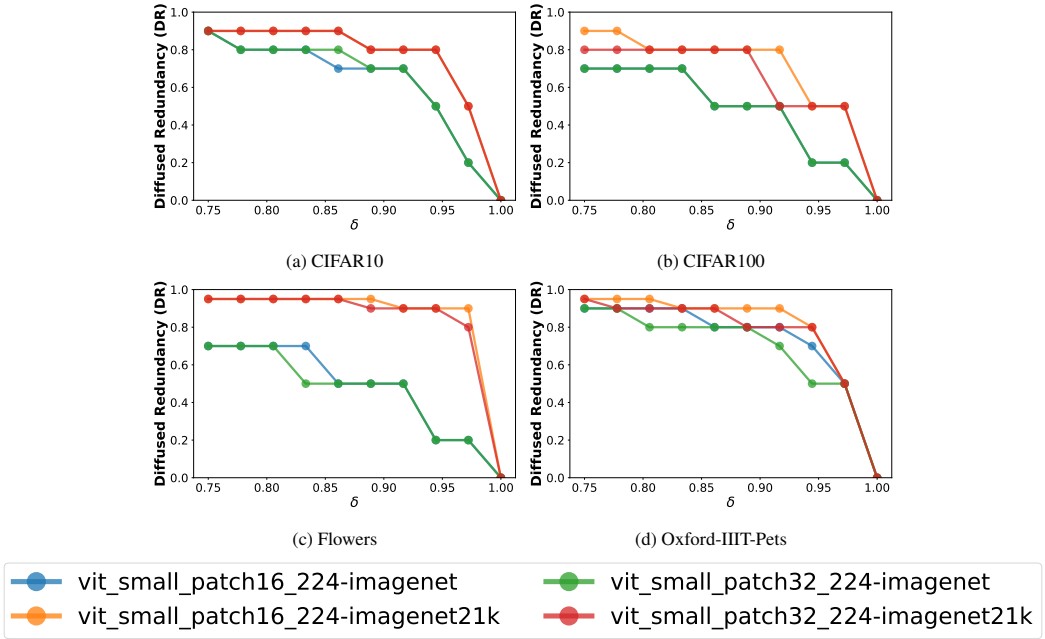

Figure 16: **[Comparison Across Upstream Datasets]** We see that degree of diffused redundancy depends a great deal on the upstream training dataset, in particular models trained on ImageNet21k exhibit a higher degree of diffused redundanacy, although the differences in the degree of diffused redundanacy are downstream task dependent

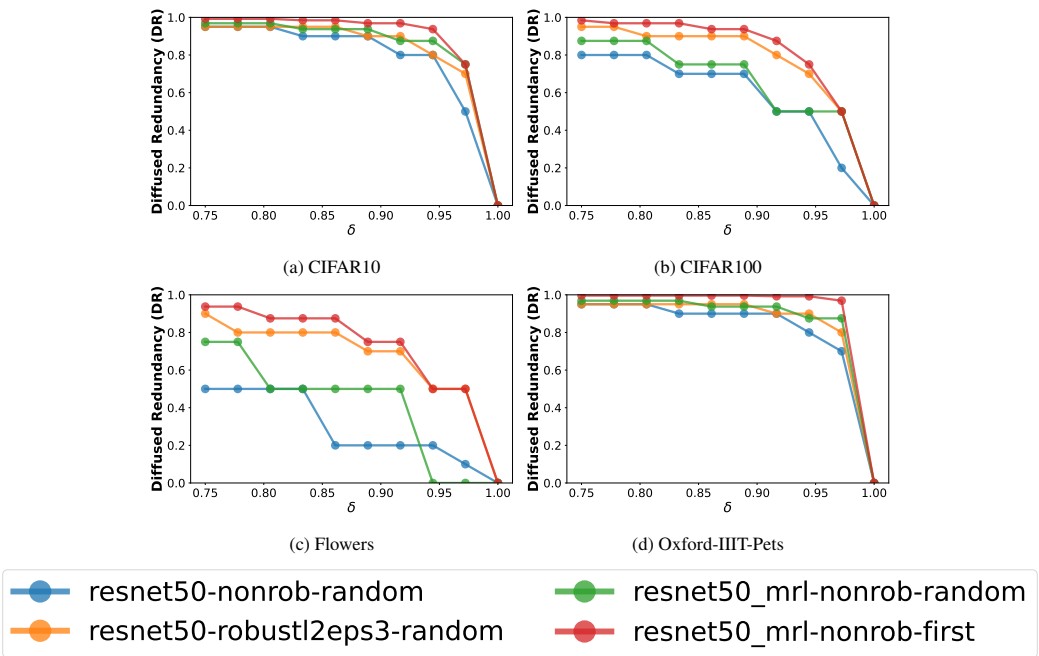

Figure 17: **[Comparison of Diffused Redundancy in MRL vs other losses]** Here we compare ResNet50 trained using multiple losses including MRL [26]. Red line shows results for part of the representation explicitly optimized in MRL, whereas green line shows results for parts that are picked randomly from the same representation. Even the MRL model shows a significant amount of diffused redundancy despite being explicitly trained to instead have structured redundancy. This figure shows diffused redundancy (DR) for all plots in Figure 7.

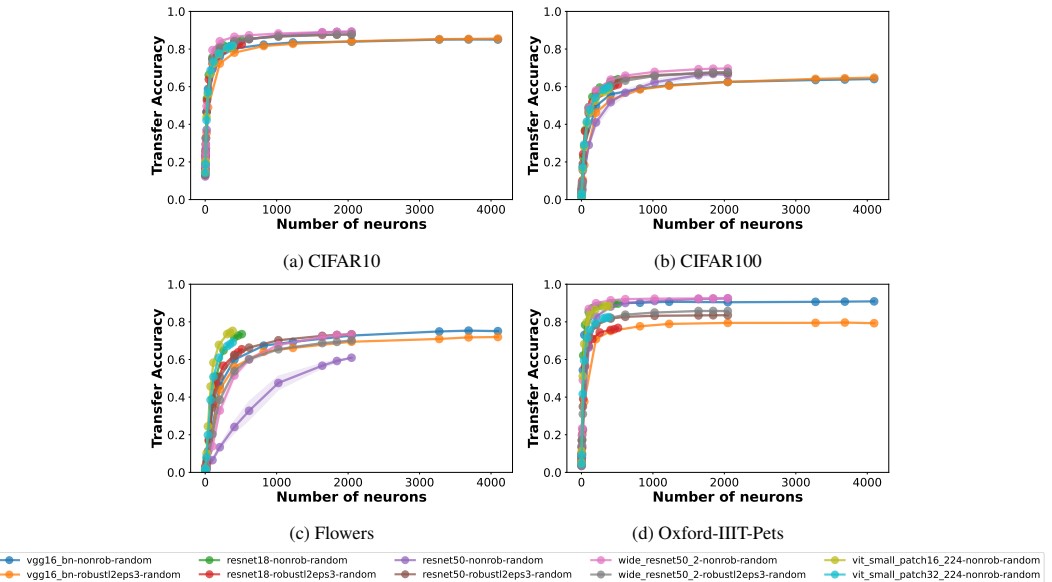

(a) CIFAR10      (b) CIFAR100

(c) Flowers      (d) Oxford-IIIT-Pets

Figure 18: **[Comparisons Across Architectures For Downstream Task Accuracy]** This shows the same plots as Figure 4, except showing absolute number of neurons on the x-axis

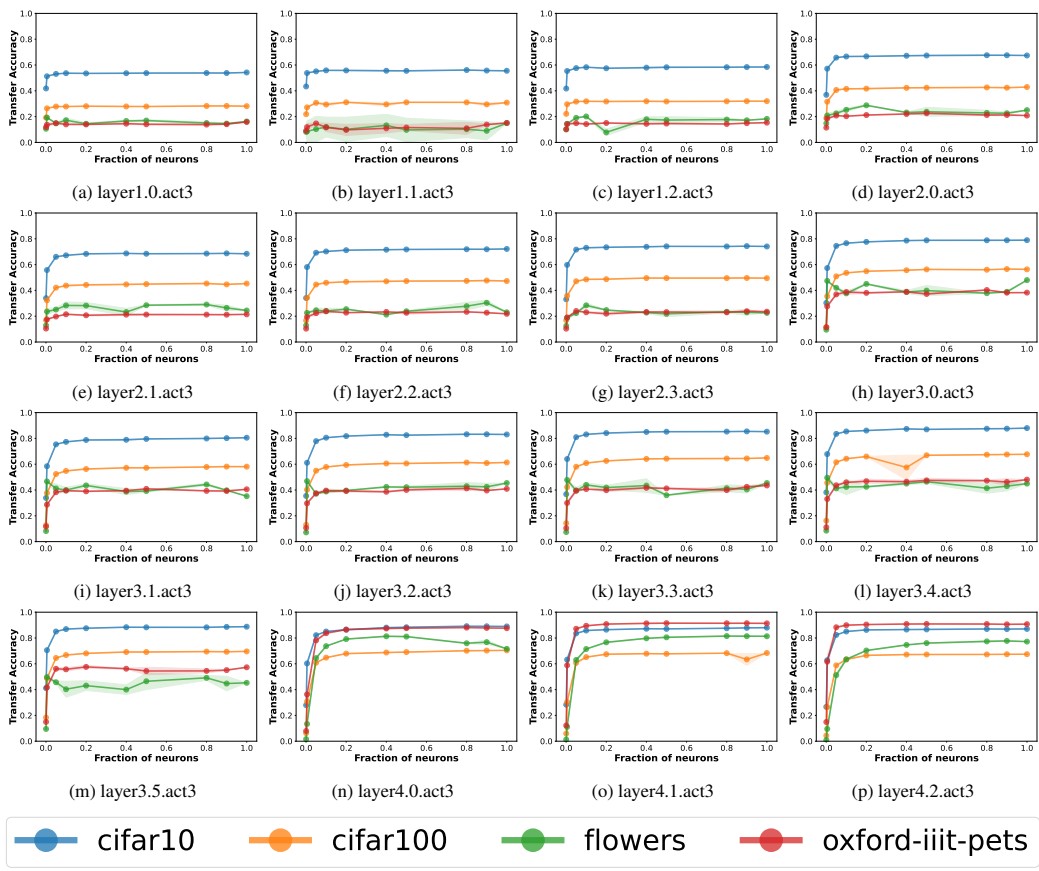

(a) layer1.0.act3   (b) layer1.1.act3   (c) layer1.2.act3   (d) layer2.0.act3

(e) layer2.1.act3   (f) layer2.2.act3   (g) layer2.3.act3   (h) layer3.0.act3

(i) layer3.1.act3   (j) layer3.2.act3   (k) layer3.3.act3   (l) layer3.4.act3

(m) layer3.5.act3   (n) layer4.0.act3   (o) layer4.1.act3   (p) layer4.2.act3

Figure 19: **[Middle Layers; ResNet50, trained with CrossEntropy loss on ImageNet1k]** We see that as we go deeper in the network, accuracy progressively increases. We see even middle layers exhibit diffused redundancy, and accuracy plateaus very quickly for earlier layers. `layerX.Y.act3` refers to the $Yth$ residual connection in the $Xth$ ResNet block and act3 indicates that we're taking the value after the activation (ReLU) has been applied.

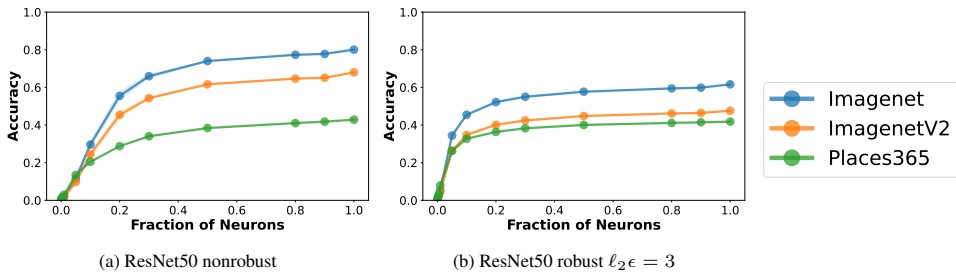

(a) ResNet50 nonrobust

(b) ResNet50 robust $\ell_2\epsilon = 3$

Figure 20: **[Performance on ImageNet1k, ImageNetV2, and Places365]** We check for the performance of randomly chosen subsets of neurons on harder tasks like ImageNet1k, ImageNetV2, and Places365. We find that diffused redundancy holds for all these harder tasks as well. Additionally, we see that randomly dropping neurons still preserves the accuracy gap between ImageNet1k and ImageNetV2.

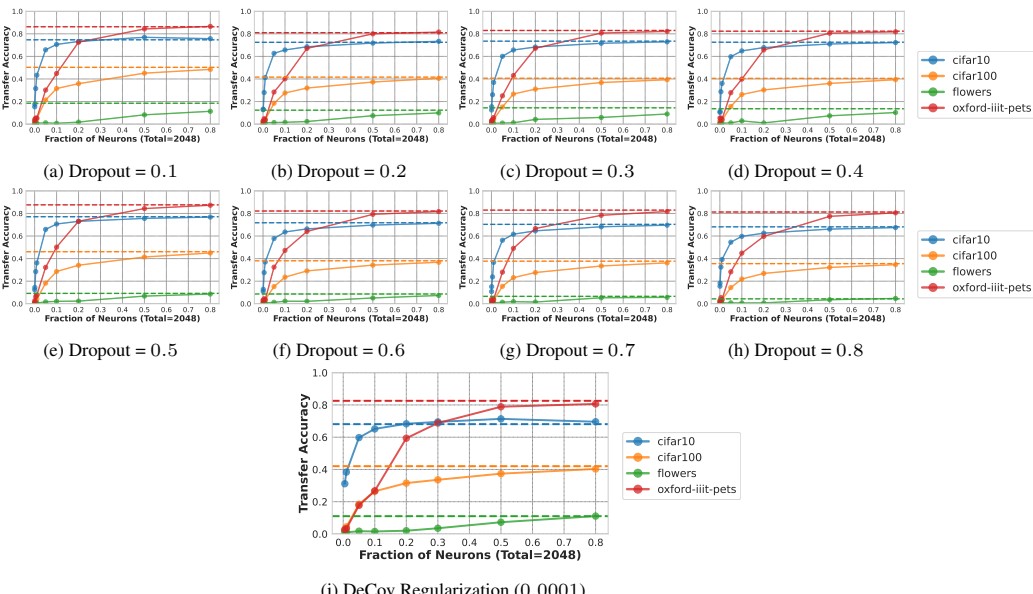

(a) Dropout = 0.1

(b) Dropout = 0.2

(c) Dropout = 0.3

(d) Dropout = 0.4

(e) Dropout = 0.5

(f) Dropout = 0.6

(g) Dropout = 0.7

(h) Dropout = 0.8

(i) DeCov Regularization (0.0001)

Figure 21: **[Dropout and DeCov regularizer's effect on Diffused Redundancy]**

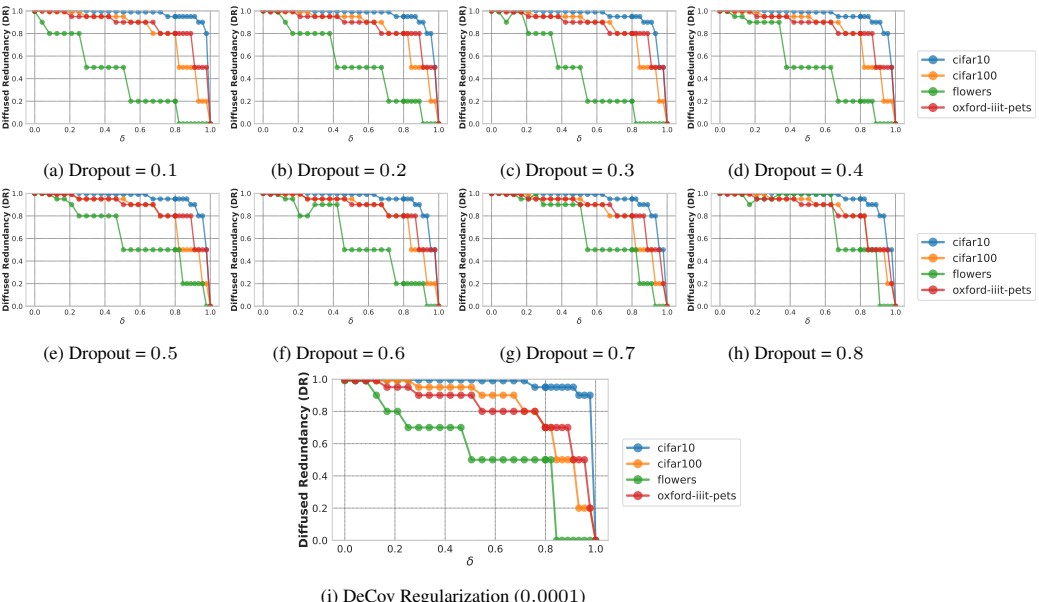

(i) DeCov Regularization (0.0001)

Figure 22: **[Dropout and DeCov regularizer's effect on Diffused Redundancy]** Same results as Figure 21, but showing $DR$ estimates (Eq 1). Lines that are more towards the right (*i.e.* more on the "outside") mean they exhibit more diffused redundancy.

