# OpenReview forum: "Diffused Redundancy in Pre-trained Representations"
_NeurIPS.cc/2023/Conference — NeurIPS 2023 poster_

### Official Review · Reviewer_1gmQ · 2023-07-04

**Soundness:** 3 good
**Presentation:** 4 excellent
**Contribution:** 3 good
**Rating:** 6
**Confidence:** 3

**Summary:**

Authors argue that pre-trained representations exhibit a degree of "diffused redundancy", i.e. the accuracy on downstream tasks with linear probing is similar when we use the whole layer compared to a randomly subset of neurons. This claim is validated using representations pre-trained using ImageNet-1k and Imagent-21k and probe these representations on downstream tasks like CIFAR-10 and CIFAR100. The authors look at different factors that influence diffused redundancy like regularization or training loss--including variants like Matryoshka learning or adversarial training. Finally, they evaluate the potential impact on different class accuracies to understand the impact of exploiting diffused redundancy.


**Strengths:**

The authors identify a new phenomenon that seems to occur when we pre-train representations on large datasets.  The manuscript methodically studies diffused redundancy on a number of different downstream tasks, regularization mechanisms, layer width and loss functions and identify that it occurs across all these scenario but to different extents. They also observe that diffused redundancy increases with adversarial training which is interesting and find that the class accuracies are impacted unevenly if we use a subset of the neurons. Overall, I believe that this is an interesting finding with no complete explanation for when and why it happens.


**Weaknesses:**

**Does diffused redundancy occur for any random subset?**

The authors claim diffused redundancy holds for *any* subset of neurons (L5 and L53). However, this claim is only validated by averaging over 5 subsets of neurons (L174). Is it possible that the redundancy is true for only some subset of neurons? For example, if we consider the representation projected onto the last $k$ principal components, does the diffused redundancy observation still hold? Alternately, we could select $k$ neurons with the least variance and test if the linear probe accuracy is similar. I believe that the claim, that this holds for *any* subset of neurons, is not sufficiently validated.

If I understand the results correctly, there exists an alternative explanation that isn't completely ruled out. As an alternative, let us assume that most of the information is diffused among an *informative subset* of neurons. Since the experiments consider only 5 random subsets of neurons, it is possible that each random subset contains some neurons from this informative subset. Hence the information may not be diffused among *all* neurons in the final layer but only among some of them.

On a related theme, in Figure 2b, the PCA projected representations has an accuracy that is higher by as much as 20% compared to a random subset of neurons. However in L56, the authors claim that the representation performs as well as PCA. Am I misinterpreting these plots? This seems to indicate that most of the information exists in a smaller subset of neurons.

**Is this specific to pre-training on ImageNet ?**

I also wonder if these observations are specific to models pre-trained on ImageNet and it would be nice to investigate this further. How prevalent is this phenomenon and does it occur if we pre-train models on Places-365 or iMaterialist? I think another important experiment is to pre-train models on smaller datasets with very few classes (like CIFAR10) and see if diffused redundancy occurs in this scenario.

**Definition of diffused redundancy**

I also have some minor concerns regarding the definition of diffused redundancy. The definition (Eqn. 1), requires the accuracy averaged over different subsets of neurons ($M_f$) to be greater than some threshold $\delta$. But this doesn't require *every* subset of neurons to have high accuracy. Maybe, a more appropriate definition is to require all subsets to have accuracy greater than some threshold if that is the central claim of the paper.


**Questions:**

The abstract (L17) claims that "entire layers might not be necessary". The authors could perhaps clarify that this is only true for the final layer, since their experiments are restricted to this layer (which doesn't discredit the importance of their findings).

In Figure 1(b) and Figure 1(d) the diffused redundancy plot seems to take only 7 different values which seems rather strange to me. Could the authors explain why this is the case? How many different $D_f$ sets are considered when estimating the diffused redundancy?

Can these observations be extended to random projections of representations? Selecting a random subset of neurons can be viewed as looking at restricted class of random projections.


**Limitations:**

The authors have studied diffused redundancy under different settings and have adequately addresses limitations. Section 4 also addresses some potential limitations if future methods were to exploit diffused redundancy.

---

> ### Author Rebuttal · Authors · 2023-08-10
>
> We thank the reviewer for their positive feedback and excellent suggestions! We answer their concerns below:
>
> ### [W#1 comparison to last $k\%$ principal components]
> Thank you for this suggestion! We’ve added these results in the response PDF (Fig 4). We see that projections onto the lower PCA components perform significantly worse than both top PCA and randomly chosen neurons (our work). This suggests that randomly sampling neurons leads to capturing features that are more akin to projections on the top (high variance) principal components, rather than the projections on the lower (low variance) principal components. We agree with you that as stated, the claim that diffused redundancy holds for *any* subset of neurons is not entirely accurate and we will reword it to say that there are many subsets that can perform as well as the entire layer.
>
> ### [W#2 what if we’re picking from an informative subset]
> This is a great point! We believe this is compatible with diffused redundancy and it’s highly likely that a random subset picks neurons from an “informative” set of neurons. In fact, we see this in Figs 2c and 2d where the performance of diffused redundancy starts to catch up with PCA’s performance as we increase the fraction of neurons sampled. Finding this exact subset of informative neurons is not straightforward, but we show that simple random sampling gets us a long way.
>
> ### [W#3 PCA vs Diffused Redundancy for lower value of $k\%$]
> We agree with the reviewer’s observation that PCA tends to be much better than diffused redundancy for very small values of $k$, but diffused redundancy mostly catches up by $k = 20\%$. We will add this nuance to the paper.
>
> ### [W#4 Other pre-training datasets]
> ImageNet models have been widely used for a variety of tasks. In addition to ImageNet1k, we also added results for ImageNet21k, which is a significantly larger version of ImageNet with weakly labeled data. We see that these models exhibit even high diffused redundancy, thus, at first glance this seems to be a more general phenomenon, not restricted to ImageNet1k.
>
> ### [W#5 worst case accuracy for Eq 1]
> Thank you for this suggestion! We agree the worst-case accuracy is the ideal formulation, especially with the way we framed this as working for “any subset”. However, there are two issues for which we prefer the current formulation that considers the mean when empirically estimating diffused redundancy: (1) in order to compute worst-case accuracy empirically, we need to run this over many more random seeds, which becomes computationally expensive, and (2) worst case can be a very pessimistic estimate. For these reasons, we will re-word our claims about “any randomly chosen subset can achieve similar performance as the entire layer” to “there exist many randomly chosen subsets that can achieve similar performance as the entire layer”.
>
> ### [W#6 results applicable only to penultimate layer]
> We have added results for other intermediate layers in the response PDF (Fig 1) and we see that diffused redundancy also exists at these layers. However, we agree that all of our other analyses are on the penultimate layer, and we will clarify this in the paper.
>
> ### [W#7 only 7 values for DR in Fig 1b and 1d]
> In practice, we calculate diffused redundancy (DR) by attaching linear probes to different fractions of randomly chosen neurons from a given layer (here penultimate layer) and finetuning just this probe on the downstream dataset. In our experiments, we chose 10 different values of this fraction and hence (1b actually has 9 distinct values) we see discrete jumps in the plot. We added more fractions in our analysis and now in Fig 3 of the response PDF, we see that DR takes many more values. Generally, as we add more fractions, we should expect to see a smoother curve for DR plots.
>
> ### [W#8 random projections vs random neurons]
> This is a very interesting suggestion! We ran this experiment and report the results in Fig 4 of the attached response PDF. Interestingly we find that diffused redundancy does significantly better than random projections. This indicates that the “constrained” projection offered by diffused redundancy is crucial in achieving good downstream performance.

---

> > ### Comment · Reviewer_1gmQ · 2023-08-12
> > **Thank you for the response**
> >
> > Thank you for the detailed response and for running new experiments. The authors have addresses most of my concerns and I have increased my score. I find the results of the new experiment using random projections (W#8) to be interesting.
> >
> > My only remaining question is regarding the prevalence of this phenomenon. Imagenet-1k/21k are somewhat similar and I wonder if this occurs if the pre-training dataset was a small dataset like CIFAR-10 or very different large dataset like Places-365/iMaterialist. Nevertheless, the paper presents many interesting results.

---

> > > ### Author Response · Authors · 2023-08-12
> > > **Thank you for your prompt response!**
> > >
> > > Thank you for engaging with our rebuttal!
> > >
> > > We agree pretraining on diverse datasets would be a valuable addition to the paper. We could not finish pre-training in time for the response, but we hope to get back to you with some numbers before the end of the discussion period!

---

### Official Review · Reviewer_yCaj · 2023-07-04

**Soundness:** 2 fair
**Presentation:** 3 good
**Contribution:** 2 fair
**Rating:** 6
**Confidence:** 4

**Summary:**

The paper empirically investigates the various factors that affect how the features of a pretrained representations are encoded. They find that the features exhibit a degree of diffuse redundancy and randomly chosen subsets of the features in the penultimate layer have similar accuracy on downstream tasks and score high in common similarity metrics. They find that the loss and datasets used to pretrain the model affect the number of features required to achieve most of the performance on a given dataset. They show potential applications for efficient transfer learning and implications for unequal performance across classes when neurons are removed.

**Strengths:**

- Using adversarial training to create more diffuse feature representations is an interesting observation and potentially useful for efficient retrieval or efficient inference when target datasets have many classes. In general, understanding how adversarial training affects the feature space of a pre-trained model is an interesting direction.
- The experiments are relatively thorough with multiple architectures and datasets.
- The paper is well written and organized.

**Weaknesses:**

- The applications and implications of the diffused redundancy hypothesis given in the paper are not not well motivated, or lacking experimental evidence.

  - One of the major stated contributions is decreased training time on downstream tasks. I think there are a few weaknesses in this argument. First, from my understanding, to determine the optimal number of neurons to drop for a given target dataset, you would need to perform multiple training runs with varying levels of neurons being dropped. Searching for this hyper-parameter would require additional FLOPS rather than reduce them. Second, I didn’t see analysis in the main paper or appendix about how many FLOPS are saved by using the reduced-length feature vector. Unless most of the FLOPS are in the final layer, the reduction in compute during training will be minor. Similar works, such as Slimmable Neural Networks, reduce parameters throughout the whole network for the purpose of efficiency, but at inference. Third, for most downstream datasets, fine-tuning is computationally cheap as the dataset and number of training epochs are small. Most of the work on efficiency is focused on pre-training or inference.

  - The other stated implication of the diffused redundancy hypothesis is the accuracy inequality between classes when reducing length of the feature vector. The experiments in figure 7 don’t rule out the possibility that the gini-coefficient is merely correlated with the accuracy. It would be more convincing to see additional models on the scatter plot in figure 7 to show that this isn’t the case, and dropping neurons in the final layer are the main cause of the increase of the gini- coefficient

- In the abstract it is stated that: “We find that learned representations in a given layer exhibit a degree of diffuse redundancy”, but from what I understand, the main experimental findings are only for the penultimate layer, so this claim is misleading.

- CIFAR, OxfordPets, and Flowers are relatively small and less diverse than the other datasets in the appendix (ImageNetV2 and Places365). It would have been useful to see more analysis of how diversity and scale affect the conclusions.

**Questions:**

- For the adversarial training experiment, is the test set the same as standard training? (i.e. are the adversarial perturbations only done on the train split.)

- The keys in the legend in figure 7 are unclear. For example, what does “nonrob” or “first” mean exactly?

- For figure 5 it would be easier to draw conclusions if the x-axis was in length of the feature vector rather than fraction of used neurons so that models could be compared. For example, 20% of a model with a 256-length feature vector isn’t comparable to 20% of a model with a 64-length feature vector. If I wanted to know the best model to use for a given feature vector length, it would be difficult to see from the figure.



**Limitations:**

The authors adequately address the limitations of their work. They discuss how removing features from the feature vector could unequally affect the class accuracy. I think further experiments are needed to convincingly show that dropping features is the root cause of this phenomenon, but the authors bring up an interesting limitation of shortening the feature vector.

---

> ### Author Rebuttal · Authors · 2023-08-10
>
> We thank the reviewer for their valuable comments! We address their concerns below:
>
> ### [W#1 decreased downstream training time]
> We would like to emphasize that our goal is not model optimization. There has been long-standing research on structured pruning (He et al.[1]; Li et al.,[2]) that achieve optimizations (in terms of FLOPS, memory, inference times vs drop in performance) which are way better than randomly dropping features (our work). Our goal, however, is to point to an intriguing property of learned features in commonly used pre-trained DNNs. We tried to make this clear in the Introduction and hence we do not mention model optimization or reduction in training FLOPS as our contribution. We would also like to emphasize that one can still use diffused redundancy in conjunction with structured pruning methods (since these methods act on initial convolution layers) to get a model that is compact overall. One can also pre-compute the parameters that are best for downstream transfer and repeatedly use them, thus not having to run the computation of diffused redundancy repeatedly. However, such analyses are outside the scope of this paper and are very interesting directions for future work!
>
>
> ### [W#2 gini being correlated to accuracy]
> This is a good point, thank you for raising this! We’ve added gini plots for more models in the response PDF (Fig 2) and see that all models follow a similar trend of increasing gini as we drop neurons. Additionally, to further investigate your point, we’ve added the accuracy distributions over CIFAR10 classes (other datasets have many classes and thus could not fit into the response PDF, but we see similar trends) as a function of fraction of neurons (due to space constraints we could only include one model, but see similar trends throughout). For the low accuracy regime (left of the x-axis in Fig 1 in the response PDF, lower fractions), the heatmap shows that a high gini coefficient is clearly due to homogeneity in predictions – the model predominantly predicts a few classes and hence falters heavily on other classes. We see that this homogeneity only appears as we gradually drop neurons, and thus can conclude that the increase in gini coefficient is primarily due to dropping neurons.
>
>
> ### [W#3 all results are only for penultimate layer]
> We have added results for other layers in the response PDF (Fig 1) and see that a similar trend of diffused redundancy holds there as well. Regardless, we agree that the majority of our results are on the penultimate layer and will tone down this claim in the paper.
>
>
> ### [W#4 larger and diverse datasets]
> We have results on ImageNet1k, ImageNetV2, and Places365, all of which are not only more diverse but are also orders of magnitude larger in size. Additionally, ImageNetV2 represents an interesting case of a distribution shift. These results are in Appendix E. We see that we still observe diffused redundancy on these datasets, despite them being much “harder” than the other small datasets we consider. We can move the discussion of these results to the main paper.
>
>
> ### [Q#1 accuracy for robust models]
> Yes, all accuracies reported are “clean” test accuracies. We will make this clear in the paper.
>
>
> ### [Q#2 legend in Fig 7]
> “First” means we take the first k neurons in the penultimate layer and “nonrob” refers to the non-robust model (i.e., model trained with the usual CrossEntropy loss). We explain this in L283-288 in the paper. We will also make it clear in the figure caption.
>
>
> ### [Q#3 Fig 5 x-axis labels]
> The goal of this analysis is not to know the best model for a given feature length but rather to show how diffused redundancy changes as we shrink the base model’s feature vector length. Since diffused redundancy is measured in terms of the fraction of neurons in a given layer, we showed plots with fractions on the x-axis. We already have the result for feature vector length in the appendix in Fig 13 for a different experiment, and see how this can sometimes be misleading since the nature of learned features varies quite a lot based on how many neurons we have in the layer. Regardless, we can certainly add the same figure with the number of neurons on the x-axis in the final version (we’re out of space in the response PDF), however, we’d kindly ask the reviewer to not hold our paper to a lower score due to this trivial change.
>
>
> [1] (Li et al., 2016) Pruning filters for efficient convnets
>
> [2] (He et al., 2017) Channel pruning for accelerating very deep neural networks

---

> > ### Comment · Reviewer_yCaj · 2023-08-13
> > **Response to Author Rebuttal**
> >
> > I thank the authors for their response. My concerns have largely been addressed. I think the implications and applications of the diffused redundancy hypothesis are still weak, however practical implications are not a stated contribution. I will raise my score to weak accept. I think the paper proposes and analyzes an interesting phenomenon, though the impact is unclear.

---

### Official Review · Reviewer_USaP · 2023-07-06

**Soundness:** 4 excellent
**Presentation:** 4 excellent
**Contribution:** 3 good
**Rating:** 7
**Confidence:** 4

**Summary:**

The authors introduce the concept of `diffused redundancy', which is the idea that a random sample of neurons reproduces the performance of the whole layer in many downstream tasks. The authors support the existence of this phenomenon with experiments on a variety of architectures pre-trained on ImageNet and explain it with the similarity between these random samples past a certain threshold size. The observed phenomenon is argued to be an 'inevitable' property of wide deep networks, as it seems to persist even when the training method should avoid it. Finally, the authors discuss the implications of diffused redundancy for fairness.

**Strengths:**

The work proposes a novel property of the hidden representations of deep neural networks which furthers our understanding of such representations. This new hypothesis is accurately framed within the existing literature and convincingly backed up by extensive numerical experiments. Overall, the manuscript is well-written and easy to follow.

**Weaknesses:**

I think this paper would benefit from a comparison of the diffused redundancy hypothesis with neural collapse (Papyan et al, 2020). In simple terms, if the neurons of the last layer are clustered into a number of groups much smaller than the width, then there will naturally be a threshold size $k$ such that if I pick $k$ neurons at random I will pick at least one from each cluster with large probability. If the number of clusters is taken as the number of classes, this would lead to a prediction for the behaviour of DR. This prediction would be the same, in principle, for different downstream tasks, but maybe this difference is only due to different downstream tasks having different classes. If true, this difference could be eliminated with an appropriate normalisation (see question 1 below).

Another criticism concerns the size of the figures. I would recommend the authors use the extra page for increasing the size of the figures.


**Questions:**

1. In line 186, the authors mention that some tasks might require more neurons simply because they have more classes. Would it be possible to find a normalised measure that takes this effect into account?

2. In Figure 2, panels (c) and (d), what is the cause of the discrepancy between dashed and solid curves for a low fraction of neurons?

3. In section 3.2, is the dependency on the width simply due to the fact that there are not enough neurons? For instance, if the fraction of random neurons required to keep performance high with width 10^3 is 10%, then of course there cannot be diffused redundancy for width 10^2. If this is the case, then I would personally demote the discussion on the width to the appendix, either to make more space for the figures or to make space for the methods that decorrelate neurons, which might be more interesting for the general audience.

Few additional comments/suggestions:

4. I believe that the related works section should also discuss the neural collapse phenomenon.

5. It would be helpful, for the sake of clarity, to have a definition of DR in words (i.e. $1$ minus the fraction of neurons required to have at least $\delta$ accuracy in downstream tasks).

**Limitations:**

the authors have adequately addressed the limitations and potential negative societal impact of their work.

---

> ### Author Rebuttal · Authors · 2023-08-10
>
> We thank the reviewer for their positive feedback and great suggestions! We address their concerns below:
>
> ### [W#1 neural collapse]
> Thank you for pointing out this interesting connection! We agree that diffused redundancy and neural collapse go hand-in-hand, and it would be especially interesting to see if diffused redundancy still holds before and around the terminal phase of training (ie before final layer activation has collapsed onto a much smaller subspace). If it’s indeed the case that neural collapse leads to diffused redundancy then we should likely not observe any diffused redundancy before the terminal phase. It will be very interesting future work to look into this! We will update the paper to include a discussion with neural collapse.
>
> ### [Q#1 normalized measure]
> Computing the current measure, as it’s stated in our paper, does not depend on the number of classes in a downstream task. This is because we consider accuracy for downstream task performance. We can further consider balanced accuracy to eliminate any effects of classes. However, the point we wanted to make in our empirical experiments was that for some datasets we find we need more neurons from the penultimate layer to achieve same performance as the entire layer. Interestingly, these also happen to be datasets with more number of classes (CIFAR100 and Flowers in our experiments).
>
> ### [Q#2 discrepancy between dashed and solid lines]
> Dashed lines show the performance of PCA features whereas solid lines show. We see that for a lower number of neurons, PCA features perform significantly better than randomly choosing neurons and hence we see this discrepancy. Once hit around $k = 20\%$ of neurons, we see that the solid line (diffused redundancy) catches up with the dotted line (PCA).
>
> ### [Q#3 width]
> Diffused redundancy depends quite a bit on the architecture that was optimized during pre-training. We see that even though with a model like VGG16 (4096 neurons) one could discard 95% neurons (keeping ~200 neurons) without taking a hit on accuracy, one can still discard more than 70% neurons on a much smaller ViT model (retaining only about 115 neurons). While across architectures these differences could be happening due to a variety of other factors, in this section we wanted to carefully test this on the same architecture with the same pre-training data, by just varying the length of the penultimate layer. We can move this to the Appendix to make space for other suggestions!
>
> ### [Clarity]
> We will indeed use the extra page for increasing the figure size, thank you for the suggestion! We also agree that an intuitive definition of DR will aid readability and will add it to the paper.

---

> > ### Comment · Reviewer_USaP · 2023-08-11
> >
> > Thanks for the response.
> >
> > I did not quite get the answer on the normalised measure. I understand that the measure itself does not depend explicitly on the number of classes, but surely there seems to be an effect due to downstream tasks having more or less classes. What I would like to know is whether this effect is entirely due to the downstream task having more or fewer classes, and hence it can be normalised somehow by using a measure which does depend on the number of classes, or not. I hope the question is clearer this time!
> >
> > regarding comparison with neural collapse, can you provide an estimate of how many neurons can be discarded without affecting accuracy under the neural collapse assumption (even if you might still not be exactly in the collapsed phase)?

---

> > > ### Author Response · Authors · 2023-08-11
> > > **Thank you for your prompt response!**
> > >
> > > We appreciate the engagement with our rebuttal.
> > >
> > > [Re: normalization and number of classes] Apologies for misunderstanding your comment about relation to the number of classes and thanks for the clarification. The effect certainly seems to depend somewhat on the number of classes when we measure downstream accuracy. However, the way we define DR in equation 1 also allows us to choose a separate task $T$ to measure DR, so we can also choose a task (for the same set of data points) that does not take any class information into account. We present preliminary results in the paper for the task of representation similarity, i.e., we calculate representation similarity using CKA [1] between randomly picked neurons and the entire layer. This task does not depend at all on any information about classes and thus, results cannot be affected by class information at all. These results are in Figures 1e and 1f -- here we see essentially no difference between any of the four datasets. This indicates that one can "normalize" the effect of number of classes on the measure of DR by choosing a task that doesn't depend on class information at all. We'll be sure to highlight this in the revised version of the paper and thank you for having this discussion. We'd also be open to hearing any other suggestions you may have about normalizing the effect of classes!
> > >
> > > [Re: Neural Collapse] If we assume neural collapse, then as per Papyan et al., *cross-example within-class variability of last-layer training activations collapses to zero, as the individual activations themselves collapse to their class means*. Thus, this means all training points collapse to $K$ points in the penultimate layer, where $K$ is the number of classes. This further implies that we need just enough information in the final layer activations to be able to represent $K$ different points. If we assume each neuron takes a binary value then, in the best case, we need $\approx log_2K$ neurons in the penultimate layer to perform the given classification task. Given that the neural network is not explicitly optimized to use a smaller representation, $log_2K$ can only happen in the best case, and the usual pick of neurons we get with random sampling will likely require more neurons than this. We see this in our experiments too, eg in Fig 1a in the paper where the curves would rise close to full layer accuracy much more sharply if we had a perfectly collapsed representation and we got the best case picks of neurons, but in reality, it takes many more neurons to achieve close to full layer accuracy than $log_2K$.
> > >
> > > Please let us know if these answer your concerns!
> > >
> > > [1] Similarity of Neural Network Representations Revisited. *Kornblith et al* https://arxiv.org/pdf/1905.00414.pdf

---

### Official Review · Reviewer_qvdG · 2023-07-10

**Soundness:** 2 fair
**Presentation:** 2 fair
**Contribution:** 2 fair
**Rating:** 5
**Confidence:** 3

**Summary:**

The paper proposes investigating the emergence of redundant representations in pre-trained models. The setup is pretraining of various architectures (VIT, vgg, resnet) on Imagenet1k. The study pertains to how much of the representation is redundant and this is estimated by dropping features and finetuning on smaller datasets (cifar 10, 100, flowers and Pets). The paper proposes a measure of redundancy based on centred kernel alignment. The experiments show that this is true under this measure for random subsets of a certain size as long as the model has enough parameters. They also show that there is an apparent drop in fairness that would have to be addressed if features are to be pruned in the proposed way for some datasets like pets and flowers.

**Strengths:**

* It is an interesting phenomenon to discover and a good study.
* The paper is generally well written.
* The comparison across models and losses is interesting and may ultimately lead to improvements in model training and or design.

**Weaknesses:**

* Not sure how this is going to impact practical aspects of ML in the future. In particular not sure how meaningful the values of CKA are here. The measure (1) seems sensible though.
* I think section 2.2 comes short of the state goal of explaining why any random subset works. It explains that this is happening but not why this is the result of the optimization.
* I find the paragraph describing the "correlation" between the pruning and PCA confusing and misleading.

**Questions:**

* I am not sure how meaningful fig 7 is if the fairness is computed on the same labels as the quality. Do I understand correctly that the claim is Gini for say cifar 10 drops less than accuracy meaning it is dropped uniformly but pet's and flowers the Gini index goes up more quickly with accuracy thus is less fair.

**Limitations:**

none.

---

> ### Author Rebuttal · Authors · 2023-08-10
>
> We thank the reviewer for their comments. We address their concerns below:
>
> ### [W#3 confusion in PCA vs Diffused Redundancy results]
> We believe the reviewer is referring to L225-235, and associated figures 2c and 2d. The goal of this analysis is to compare diffused redundancy (our work) with PCA. In PCA the goal is to project the penultimate layer onto the first $k\%$ principal components and compare how this performs to randomly choosing $k\%$ of neurons in the final layer. We see that apart from very low values of $k$, performance of random neurons closely follows the performance of PCA. Please let us know if you’d like any further clarification or if you have any suggestions for us to improve the presentation of these results.
>
> ### [W#2 sec 2.2 doesn’t explain why any subset works]
> We believe we do offer some intuition for what could be going on by showing that randomly choosing $k\%$ neurons capture the same amount of variance as top $k\%$ principle components, for moderate values of $k$. We agree this does not answer the question of why gradient descent chooses solutions like this over other solutions that utilize the entire layer. However, we believe this warrants a separate investigation and is outside the scope of this paper.
>
> ### [Q#1 meaningfulness of fig 7 when gini is calculated over class labels]
> Yes, the reviewer’s interpretation is correct. We’ve done a deeper dive into this analysis to ensure its validity and we see that class accuracies are uniformly spread, and as we drop neurons, the accuracies become less uniform. We have added this in Fig 2 of the response PDF. However, we would like to clarify that these are toy datasets without many real-world fairness implications. Our goal was to show that such disparities can arise when using “compact” representations (such as choosing $k\%$ of random neurons) and this “fairness” aspect must be taken into account when the underlying deep neural network is being used for high-stakes scenarios (eg: facial recognition).

---

> > ### Comment · Reviewer_qvdG · 2023-08-18
> > **comment**
> >
> > I thank for authors for their work and comments. I am maintaining my rating. I think the paper warrants being accepted!

---

### Official Review · Reviewer_MR5K · 2023-07-25

**Soundness:** 3 good
**Presentation:** 3 good
**Contribution:** 3 good
**Rating:** 6
**Confidence:** 4

**Summary:**

Focusing on the (penultimate layer) representation learned by the pre-training model, this paper proposes a "diffused redundancy hypothesis" that includes two main concepts:

- Redundancy: A small subnet can achieve comparable performance with the full model.
- Diffused redundancy: The selection of the small subnet can be random.

To validate this hypothesis, this paper conducts experiments across various models, datasets, and training paradigms (e.g., standard training or adversarial training) to demonstrate the diffused redundancy phenomenon. Furthermore, this paper also investigates the possible factors that influence the degree of this phenomenon. Finally, this paper shows the trade-off between fairness (class-wise accuracy) and efficiency (pruning rate).

**Strengths:**

1. The proposed diffused redundancy hypothesis is an unrealized phenomenon and may have a strong impact on the research community.
2. This paper leverages several metrics, such as Centered Kernel Alignment (CKA), to interpret and understand this phenomenon, which is reasonable and enhances the soundness of the proposed hypothesis.
3. This paper systematically investigates the underlying mechanism and influencing factors of the diffused redundancy hypothesis.

**Weaknesses:**

1. Given that model pruning and compression techniques can achieve comparable performance with only a few neurons, the diffuse redundancy phenomenon seems to be expected and less appealing. Intuitively, a wide network inevitably results in diffuse redundancy in the representations.
2. This paper only investigates the representation on the penultimate layer and does not analyze this phenomenon in terms of middle layers.
3. This paper only conducts empirical experiments to validate the proposed hypothesis but does not provide theoretical analysis of why this phenomenon exists.
4. This paper does not provide an intuitive or theoretical understanding of the fairness-efficiency trade-off in Section 4.
5.  Line 163, Equation 1. $DR(g, T, \delta)$ is defined with the average $\frac{T(m \odot g)}{T(g)}$ for $m\in M_f $.
However, the diffuse redundancy hypothesis claims that any randomly chosen subset of neurons can achieve similar performance. Therefore, I think using the worst representation similarity instead of the average is more appropriate: $$DR(g,T,\delta)=\frac{1-\min f, \text{s.t.}\min_{m\in\mathcal{M}_f} \frac{T(m\odot g)}{T(g)}\ge \delta}{|g|}$$

**Questions:**

1. Line 178: How about $\ell_\infty$ adversarial training and other perturbation bounds with $\epsilon$?
2. Lines 312-314: This paper claims that diffused redundancy can be used to reduce training and storage costs. Will model pruning or compression techniques serve a similar function? Will carefully pruning $k$% of the neurons result in better performance than randomly dropping $k$% of neurons? ($0\le k \le 100$)

---

> ### Author Rebuttal · Authors · 2023-08-10
>
> Thank you for your careful review! We address your concerns below:
>
> ### [W#1 obvious in light of model compression]
> We draw differences between our work and model compression literature in our paper in Section 1.1. To expand on that, model compression works aim to reduce the number of parameters in the neural network, without necessarily getting rid of entire neurons. In fact, pruning a ResNet50 by using the lottery ticket hypothesis [1] results in a penultimate layer of the same size as in the dense model. The main promise of such works is to be able to deploy “big” models in resource-constrained environments (eg: edge devices) since pruned models can speed up computation by making use of sparse matrix multiplications and require less storage (if most weights are zero then all these weights do not need to be stored).
> On the other hand, our work investigates the nature of learned representations by dropping entire neurons from layers, without pruning away any parameters. Thus, we differ subtly, but quite significantly from traditional works on pruning. We would like to emphasize that our work is not obvious in light of work on pruning. The fact that even weight-pruned networks require all neurons to retain high performance should indicate that all neurons are necessary for downstream performance, but we show that this is not the case. Our work can be combined with other works on pruning, which will be an exciting avenue for future work.
> We completely agree with the reviewers’ intuition about wide networks having more diffused redundancy, however, we believe such intuitions should be empirically tested and our work provides a principled framework to do that. We also find in our experiments that WideResNet50 exhibits more diffused redundancy than ResNet50 (Fig 10, Appendix).
>
> ### [W#2 Intermediate Layers]
> One reason for focusing on the penultimate layer in the paper is that that’s what’s typically used for downstream tasks. Nonetheless, we agree with the reviewer that this will be an interesting phenomenon to test at intermediate layers as well, and hence we’ve repeated our experiments for other layers – we find that similar observations hold even for intermediate layers. See Fig 1 in the attached response PDF for the plots.
>
> ### [W#3&4 Lack of theory]
> We would like to emphasize that the goal of our paper was not to provide a theoretical explanation of this phenomenon and hence we believe this is unjustified criticism. We make it very clear in the abstract that this is an empirical paper. Theoretical analysis of this phenomenon is certainly interesting for future research. Our goal, however, is to introduce the intriguing phenomenon of diffused redundancy and investigate it thoroughly by conducting empirical analyses.
> We would also like to highlight that theoretical analysis is not trivial here, as is the case with many deep learning papers since any theoretical analysis needs to take into account many nuances of optimizing these models such as learning rate schedules, pre-training data distribution, downstream data alignment with pretraining data, architectural differences such as using self-attention vs convolutions, layernorm vs batchnorm, size of layers and so on. We do not yet have the basic tools to theoretically analyze all these phenomena, and thus turn to an empirical analysis of all these factors in our paper. Moreover, as pointed out in many debates at past ML conferences, theory usually follows after an empirical observation has been made [2]. This is also indicated in recent papers in deep learning that have introduced an interesting empirical phenomenon (Eg: The Lottery Ticket Hypothesis [1]) which have then been followed up with theoretical papers explaining the phenomenon (eg, see [3,4] with the theory behind the Lottery Ticket Hypothesis).
>
> ### [W#5 Taking worst case in Eq 1]
> This is a good point, thank you for the suggestion! We agree the worst-case accuracy is the ideal formulation, especially within our framing of “any subset works”. However, there are two issues for which we prefer the current formulation that considers the mean: (1) in order to compute worst-case accuracy empirically, we need to run this over many more random seeds, which becomes computationally expensive, and (2) worst case can be a very pessimistic estimate. For these reasons, we will re-word our claims about “any randomly chosen subset can achieve similar performance as the entire layer” to “there exist many randomly chosen subsets that can achieve similar performance as the entire layer”.
>
> ### [Q#1 $\ell_\inf$ models]
> Thanks for this suggestion! We have added results for $\ell_\inf$ robust models, please see Fig 3 in the attached response PDF. We see that $\ell_2$ robust models have slightly higher diffused redundancy than $\ell_\inf$ models, which in turn have higher redundancy than nonrobust models.
>
> ### [Q#2 Carefully selecting $k\%$ neurons]
> We have already included results for PCA, which carefully chooses dimensions to project the final layer onto. These are shown in Figures 2c and 2d and are discussed in Section 2.2. We have also added more results for different strategies for choosing $k\%$ neurons in the response PDF (Fig 4) based on reviewer 1gmQ’s suggestions. All of these results show that randomly selected neurons have unexpectedly high predictive power.
>
>
> [1] The Lottery Ticket Hypothesis: Finding Sparse, Trainable Neural Networks by Frankle and Carbin https://arxiv.org/abs/1803.03635
>
> [2] My take on Ali Rahimi's "Test of Time" award talk at NIPS by Yann LeCun https://www2.isye.gatech.edu/~tzhao80/Yann_Response.pdf
>
> [3] Proving the Lottery Ticket Hypothesis: Pruning is All You Need by Malach et al. https://arxiv.org/abs/2002.00585
>
> [4] Optimal Lottery Tickets via Subset Sum: Logarithmic Over-Parameterization is Sufficient Pensia et al. https://arxiv.org/abs/2006.07990

---

> > ### Comment · Reviewer_MR5K · 2023-08-10
> >
> > Dear authors,
> >
> > Thanks for the rebuttal. I think most of my concerns has been successfully addressed. I will raise my score when review editing is allowed.
> >
> > I still have a little concern in your reply on W#5. You stated you will re-word the claim as
> > > there exist many randomly chosen subsets that can achieve similar performance as the entire layer
> >
> > However, it's still unclear exactly how may subsets can achieve this. I think you can propose a corresponding metric or further adjust this claim to address this.
> >
> > Best,

---

> > > ### Author Response · Authors · 2023-08-11
> > > **Thank you for your prompt response!**
> > >
> > > We thank you for engaging with our rebuttal!
> > >
> > > You are right, it's not exactly clear how many of these subsets exist, especially because testing out all the subsets is going to be infeasible (e.g.: for a ResNet50 for choosing 10% neurons in the final layer requires testing out ${2048 \choose 204} \approx 6.994519 \times 10^{286}$ combinations) -- for the new experiments in our rebuttal for $\ell_\inf$ models, we ran over 20 random seeds and did not observe any significant difference from results over 5 random seeds. We suspect that there are probably a few subsets that cannot transfer as well, but we have had a hard time finding them with random sampling. In light of diffused redundancy, it will be interesting to come up with a principled way to find neurons that *do not* perform well.
> > >
> > > We will add additional clarification to our claim stating that finding exactly how many subsets of neurons work well without running all possible combinations remains an open challenge for future works.

---

> > > > ### Comment · Reviewer_MR5K · 2023-08-11
> > > >
> > > > Dear authors,
> > > >
> > > > Thanks for your response. I have updated my score based on discussion above.
> > > >
> > > > Regarding the Fairness-Efficiency Tradeoffs, there also exists a robustness-fairnss tradeoff [1,2]. Is there any connection between this tradeoff in robust models and the tradeoff mentioned in your paper?
> > > >
> > > > Additionally, will adversarial training algorithms that improves robustness-fairness trade-off like CFA [2] can also improve Fairness-Efficiency Tradeoffs in efficient downstream transfer?
> > > >
> > > > Note that the theoretical insight presented in [2] may also help you intuitively explain the Fairness-Efficiency tradeoffs.
> > > >
> > > > [1] To be Robust or to be Fair: Towards Fairness in Adversarial Training. ICML
> > > >
> > > > [2] CFA: Class-wise Calibrated Fair Adversarial Training. CVPR
> > > >
> > > > Best,

---

> > > > > ### Comment · Reviewer_MR5K · 2023-08-12
> > > > > **No reply?**
> > > > >
> > > > > Dear authors,
> > > > >
> > > > > Please let me know if you face any difficulties regarding the questions above.
> > > > >
> > > > > Also, note that due to time limitations, the experimental result of the questions above is optional. You can only discuss the connection and the explanation intuitively.
> > > > >
> > > > > Best,

---

> > > > > > ### Comment · Reviewer_MR5K · 2023-08-20
> > > > > > **Still waiting for reply**
> > > > > >
> > > > > > Dear authors,
> > > > > >
> > > > > > Note that the deadline of the discussion period is approaching. If you have any comments regarding my proposed aspect, please reply promptly.
> > > > > >
> > > > > > Best,

---

> > > > > > > ### Author Response · Authors · 2023-08-20
> > > > > > > **Not sure why our response didn't reach you. Apologies and thanks for pinging us again!**
> > > > > > >
> > > > > > > Dear reviewer,
> > > > > > >
> > > > > > > We responded on 12th August 2023. We're not sure why our response didn't reach you, please accept our apology for what seems like an OpenReview error. Thank you for pinging us. We'll paste our response again, please let us know if you're able to see it:
> > > > > > >
> > > > > > >
> > > > > > > Thank you for pointing out these exciting connections! We were trying to see if we could pretrain an ImageNet model using either FRL or CFA to present some empirical results in our response. However, the training times are in the order of weeks and thus we greatly appreciate your understanding in this matter!
> > > > > > >
> > > > > > > This is nonetheless a very interesting connection at a conceptual level. Both FRL and CFA ([1] and [2] respectively) build on the main observation that optimizing for population-level robust loss in adversarial training leads to a much lower accuracy (both clean and robust) for some classes, leading to a potential fairness concern. One major difference between these works and our setup is that they do end-to-end adversarial training, i.e., the task they train on is also the task they test on. We instead operate in the transfer learning setup where we take the backbone of an adversarially trained model on ImageNet and then use it to transfer to a downstream task (we only do the standard training when transferring to a downstream task). While the observations of FRL and CFA could hold true for accuracies on ImageNet classes, it's not immediately clear whether discrepancy in ImageNet class accuracy will lead to discrepancy on downstream tasks as well. For example, in Fig 7 of our paper, for all downstream datasets, when using the full representation, we see gini coefficient to be very close to zero, indicating almost similar accuracy for all downstream classes. However, when we start to drop neurons for the robustly trained model and then transfer to downstream tasks, we sometimes see a sharper rise in the gini of a robust model than that of the usual nonrobust model. This can be seen for example for CIFAR10 in both Fig 7a in our paper and Fig 2b in the response PDF. To corroborate whether this is a direct consequence of the robustness-fairness tradeoff observed by [1] and [2], it would be very interesting to run FRL/CFA at ImageNet scale and add it to the plots in Figure 7. Intuitively, if the robustness-fairness tradeoff also affects the fairness-efficiency tradeoff presented in our paper then we'd expect the FRL/CFA trained model's gini to not rise as sharply as the usual adversarially trained model.
> > > > > > >
> > > > > > > [1] To be Robust or to be Fair: Towards Fairness in Adversarial Training. ICML
> > > > > > >
> > > > > > > [2] CFA: Class-wise Calibrated Fair Adversarial Training. CVPR

---

> > > > > > > > ### Comment · Reviewer_MR5K · 2023-08-20
> > > > > > > > **Thanks for the further response**
> > > > > > > >
> > > > > > > > Dear authors,
> > > > > > > >
> > > > > > > > Thanks for the further response. I agree that an OpenReview error may have occurred.
> > > > > > > >
> > > > > > > > Given these discussion, I would like to recommend accepting this paper and have raised my score.
> > > > > > > >
> > > > > > > > Please add these discussion (and corresponding experiments, if time permits) in the future version of this paper, which can substantial the claims on the proposed trade-off.
> > > > > > > >
> > > > > > > > Best,

---

> > > > > > > > > ### Author Response · Authors · 2023-08-20
> > > > > > > > > **Thank you for engaging!**
> > > > > > > > >
> > > > > > > > > We will add these discussions to our next revision. Thank you for your valuable suggestions!

---

### Author Rebuttal · Authors · 2023-08-10

We thank the reviewers for their time, positive comments and constructive feedback. There were many excellent points raised by the reviewers. We ran the following experiments to address their concerns:

 - Fig 1: One common comment raised by reviewers was that all our experiments were only on the penultimate layer. We additionally ran our experiment on other intermediate layers and report the results here. We present results for a ResNet50 pretrained on ImageNet1k using the standard CrossEntropy loss – we see similar trends for other models as well, but have to omit the results due to space constraints. The intermediate layers considered are characterized as activations following each residual connection within distinct ResNet blocks. We show 4 such layers, each from a different block of the ResNet. Interestingly, the results indicate the existence of diffused redundancy even within intermediate layers.

 - Fig 2: This comment was raised by reviewer yCaj. With this analysys we wish to make sure that the reported drop in gini coefficient is actually due to dropping neurons and not merely an artifact of low accuracy. We show class-wise accuracies as a function of dropping neurons, along with gini plots of other models and find that while class-wise accuracies start at (almost) uniform values, as we drop neurons, performance on some classes deteriorates more than other classes, thus showing that dropping neurons directly results in disparate performance across classes.

 - Fig 3: We added diffused redundancy (DR) results for a $\ell_\inf$ ResNet50 robust model. We also added more fractions that are used to compute Diffused Redundancy, so the curve now takes "smoother" values than shown in the paper. We find that $\ell_2$ robust models show higher diffused redundancy than $\ell_\inf$ models.

 - Fig 4: Added comparisons of DR with random projections, projections on top PCA components (high variance), and projections on lower PCA components (low variance). We see that diffused redundancy works massively better than random projections and least PCA dimensions.

---

### Decision · Program_Chairs · 2023-09-21

**Decision:**

Accept (poster)

**Comment:**

All reviewers unequivocally agreed that this submission to be accepted.